

# Gas exchange estimates in the Peruvian upwelling regime biased by multi-day near-surface stratification

Tim Fischer[1], Annette Kock[2], Damian L. Arévalo-Martínez[2], Marcus Dengler[1], Peter Brandt[1,3], and Hermann W. Bange[2]

[1]GEOMAR Helmholtz Centre for Ocean Research Kiel, Physical Oceanography, Kiel, Germany
[2]GEOMAR Helmholtz Centre for Ocean Research Kiel, Chemical Oceanography, Kiel, Germany
[3]Kiel University, Kiel, Germany

**Correspondence:** Tim Fischer (tfischer@geomar.de)

**Abstract.** The coastal upwelling regime off Peru in December 2012 showed considerable concentration gradients of dissolved nitrous oxide ($N_2O$) across the top few meters of the ocean. The gradients were predominantly downward, i.e. concentrations decreased toward the surface. Ignoring these gradients causes a systematic error in regionally integrated gas exchange estimates, when using observed concentrations at several meters below the surface as input for bulk flux parameterizations - as is routinely

practiced. Here we propose that multi-day near-surface stratification events are responsible for the observed near-surface $N_2O$ gradients, and that the gradients induce the strongest bias in gas exchange estimates at winds of about 3 to 6 m s$^{-1}$. Glider hydrographic time series reveal that events of multi-day near-surface stratification are a common feature in the study region. In the same way as shorter events of near-surface stratification (e.g. the diurnal warm layer cycle), they preferentially exist under calm to moderate wind conditions, suppress turbulent mixing, and thus lead to isolation of the top layer from the waters below

(surface trapping). Our observational data in combination with a simple gas-transfer model of the surface trapping mechanism show that multi-day near-surface stratification can produce near-surface $N_2O$ gradients comparable to observations. They further indicate that diurnal and shorter stratification cycles can only create $N_2O$ gradients that do not substantially impact emission estimates. Quantitatively, we estimate that the integrated bias for the entire Peruvian upwelling region in December 2012 represents an overestimation of the total $N_2O$ emission by about a third, if concentrations at 5 m or 10 m depth are used as

surrogate for bulk water $N_2O$ concentration. Locally, gradients exist which would cause emission overestimations by a factor of two or more. As the Peruvian upwelling region is an $N_2O$ source of global importance, and other strong $N_2O$ source regions could tend to develop multi-day near-surface stratification as well, the bias resulting from multi-day near-surface stratification may also impact global oceanic $N_2O$ emission estimates.

## 1 Introduction

This study develops its results and conclusions for the exemplary case of dissolved nitrous oxide ($N_2O$), but many aspects will also be valid for other dissolved gases, particularly for gases with similar solubility in seawater. Oceanic upwelling regimes have been increasingly recognized as strong emitters of ($N_2O$), particularly if they are in vicinity of oxygen deficient waters (Codispoti et al., 1992; Bange et al., 1996; Nevison et al., 2004; Naqvi et al., 2010; Arévalo et al., 2015). $N_2O$ is of global

importance mainly after its emission to the atmosphere, due to its strong global warming potential (Wang et al., 1976; Myhre et al., 2013) and its involvement in the depletion of stratospheric ozone (Hahn and Crutzen, 1982; Ravishankara et al., 2009). Although oceanic $N_2O$ emissions very likely constitute a major fraction of the atmospheric $N_2O$ budget, they are not well constrained (Ciais et al., 2013). This is particularly the case for upwelling regions (Nevison et al., 2004; Naqvi et al., 2010).

In order to better quantify oceanic $N_2O$ emissions, there have been several studies in the past: e.g. with global perspective (Elkins et al., 1978; Nevison et al., 1995; Suntharalingam and Sarmiento, 2000; Bianchi et al., 2012), and with particular focus on upwelling regions (Law and Owens, 1990; Nevison et al., 2004; Cornejo et al., 2007; Naqvi et al., 2010; Kock et al., 2012; Arévalo et al., 2015) because of their anticipated role as emission hotspots. What causes upwelling regimes to exhibit strong emissions is the transport of intermediate and central waters with accumulated $N_2O$ toward the surface, and the usually high

level of local biological production and remineralization. The high biological activity also includes microorganisms participating in the nitrogen cycle, which can provide an additional local $N_2O$ source (Nevison et al., 2004). The local source can intensify tremendously under low oxygen conditions. Particularly strong net accumulation of $N_2O$ is observed at locations that are peripheral to anoxic conditions (Codispoti and Christensen, 1985; Codispoti et al., 1992; Naqvi et al., 2010; Ji et al., 2015; Kock et al., 2016). This is probably due to three interacting effects of the particular oxygen conditions here: enhanced $N_2O$

production by nitrifiers and denitrifiers both working increasingly imperfect when about to pass the oxygen limits of their respective metabolism (Codispoti et al., 1992; Babbin et al., 2015), co-existence of oxidative and reductive metabolic pathways that would exclude each other in higher or lower oxygen conditions (Kalvelage et al., 2011; Lam and Kuypers, 2011) thus enabling a fast nitrogen turnover (Ward et al., 1989) including a fast $N_2O$ turnover (Codispoti and Christensen, 1985; Babbin et al., 2015), and sharp oxygen gradients and strong short-term variations of ambient oxygen conditions which guarantee that

the oxygen level of optimum $N_2O$ production is met at some fraction of time (Naqvi et al., 2000). The Peruvian upwelling regime intersects a pronounced oxygen minimum zone with a large anoxic volume fraction and a typically sharp oxycline, and thus offers best conditions for such peripheral hotspot $N_2O$ production (Kock et al., 2016).

To date, most studies that estimate regional oceanic $N_2O$ emissions from observations are based on dissolved $N_2O$ concentrations at some meters below surface (e.g., Law and Owens, 1990; Weiss et al., 1992; Rees et al., 1997; Rhee et al., 2009;

Kock et al., 2012; Farías et al., 2015; Arévalo et al., 2015). Similarly, air-sea gas exchange estimates of other gas species are also often based on measurements at some meters below surface, or 'near-surface'. Usually the chosen sample depths lie within the top 10 m of the water column. This is why for the course of this paper we define the near-surface to be the top 10 m range, even if usually 'near-surface' is a qualitative label for the upper few meters, without fixed limits. The measured concentrations are then used to calculate local air-sea gas exchange according to

$$\Phi = k_w \cdot \Delta c. \tag{1}$$

The flux density $\Phi$ across the surface is determined by the concentration difference between water and air ($\Delta c$) and a transfer velocity ($k_w$). $\Delta c$ is assumed to be well described by a measured concentration somewhere in the near-surface ($c_{ns}$) and the concentration at the immediate water surface in equilibrium with the atmosphere ($c_{eq}$, controlled by atmospheric mole fraction





and solubility). Thus it is assumed that $\Phi$ is well estimated by

$$\Phi_{ns} = k_w \cdot \Delta c_{ns} = k_w \cdot (c_{ns} - c_{eq}). \tag{2}$$

This measurement strategy is inspired by the formulation of bulk flux parameterizations, with

$$\Phi_{bulk} = k_w \cdot \Delta c_{bulk} = k_w \cdot (c_{bulk} - c_{eq}), \tag{3}$$

requiring the concentration in the 'bulk water' ($c_{bulk}$) instead of $c_{ns}$. The term 'bulk' suggests constancy of properties across a not too thin layer. $c_{bulk}$ is conventionally understood as the concentration within a layer of homogeneous concentration that immediately adjoins to the viscous boundary layer (Garbe et al., 2014). As this paper focuses on near-surface concentration gradients, we do not want to assume the guaranteed existence of a homogeneous layer down to a certain depth. Nevertheless, we keep the term $c_{bulk}$ for the concentration below the viscous boundary layer, even for the limiting case of an infinitesimally

thin homogeneous layer. $k_w$ in equation 2 is assumed to be identical to equation 3. This includes the assumptions that (i) either concentrations are expected to be homogeneous from measurement depth up to the bulk level, so that $c_{ns} = c_{bulk}$ everywhere, or (ii), $c_{ns}$ and $c_{bulk}$ are expected to differ unsystematically in space and time, so that treating measurements as if $c_{ns} = c_{bulk}$ would not result in a systematic error in regionally averaged $\Phi$.

Here we challenge these assumptions, at least for the Peruvian upwelling region, by showing that $N_2O$ gradients exist in the

topmost meters of the ocean, which are both considerable and systematic. The observed gradients are predominantly downward, i.e. $N_2O$ concentrations decrease toward the surface. This evokes a principal systematic measurement issue when assuming $c_{ns} = c_{bulk}$ (the 'Delta c sampling issue' with the use of bulk flux parameterizations). We propose a process, namely multi-day near-surface stratification, to be responsible for substantial $N_2O$ gradients in conditions typical for upwelling regions, and further support this by observations and simple model calculations. Finally, we estimate the total emission bias for the Peruvian

upwelling region in December 2012.

This study was initially motivated by an apparent mismatch between $N_2O$ emission and $N_2O$ supply to the mixed layer in the Mauritanian upwelling region (Kock et al., 2012). One of several hypotheses to reconcile this was to assume that the mismatch is caused by overestimated emissions due to the Delta c sampling issue in downward near-surface $N_2O$ gradients. Could - in principle - very shallow stratified layers that were encountered before in upwelling regions account for substantial vertical

$N_2O$ gradients and overestimated emission rates? Temporal near-surface stratification above the seasonal pycnocline has been observed since several decades (e.g., Stommel and Woodcock, 1951; Bruce and Firing, 1974; Soloviev and Vershinsky, 1982). Observations mainly from the open ocean revealed a diurnal cycle of near-surface temperature which is associated with the build-up of shallow stratification during daytime and its destruction during nighttime. The build-up of near-surface stratification is due to solar differential heating of the top few meters of the ocean, with high insolation and weak wind as important

prerequisites for strong effects (e.g., Soloviev and Lukas, 1997; Gentemann et al., 2008). This diurnal cycle of near-surface temperature and stratification ('diurnal warm layer cycle') has been extensively modeled and observed (e.g., Imberger, 1985; Price et al., 1986; Fairall et al., 1996a; Gentemann et al., 2003, 2009; Prytherch et al., 2013; Wenegrat and McPhaden, 2015). The strong stratification dampens turbulence and isolates a surface homogeneous layer from the water below ('surface trapping'


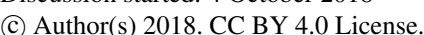

of Price et al. (1986); 'capping layer' of McNeil and Merlivat (1996); Soloviev and Lukas (1997)), such that vertical gradients of any water property can develop if supply/source and loss/sink terms differ between above and below the isolating interface. For dissolved gases, vertical gradients in the top meters due to surface trapping had been predicted (McNeil and Merlivat, 1996), and later were observed in the open ocean (Soloviev et al., 2002; Calleja et al., 2013). The two studies showed that

concentration differences of oxygen and carbon dioxide exist across the top meters of several open ocean regions, however with little average effect on gas exchange estimates. In coastal upwelling regions, there have been no reports of near-surface gas gradients. However, conditions for near-surface stratification and gradients should be more favorable here than in the oligotrophic open ocean, because of stronger near-surface light absorption in the chlorophyll enriched water, and because of the tendency of wind decreasing toward the coast (Chavez and Messié, 2009).

Typically, it is assumed that the near-surface stratification that has formed during daytime is completely eroded during nighttime through convective and shear-driven mixing, generating a diurnal cycle of near-surface stratification. Night survival of near-surface stratification would prolong the surface trapping tremendously, more than just by the additional night hours, because the pre-existing stratification next morning eases surface trapping of heat during the following daylight insolation. It thus amplifies and stabilizes near-surface stratification in a positive feedback, and makes it more unlikely that this stratification

is destroyed before the following evening. Such events extending beyond the diurnal timescale have not been explicitly investigated before, but hints for their existence can be found in reported observations of Stommel and Woodcock (1951), Stramma et al. (1986), Prytherch et al. (2013). Multi-day near-surface stratification showed up prominently during our field observations in the Peruvian upwelling region, and will be discussed as major factor responsible for substantial vertical gas gradients in section 4. The Peruvian upwelling region was chosen as suitable study site because very high $N_2O$ concentrations had been

found here already before the campaign in 2012/13 (Nevison et al., 2003; Kock et al., 2016), which then expectedly cause large vertical concentration differences that should be more easily detected with statistical significance than elsewhere.

## 2 Data and methods

### 2.1 Data overview

In the context of a ship based survey campaign from December 2012 to February 2013 in the Peruvian upwelling region,

the cruise Meteor 91 (M91, carried out within the scope of BMBF project SOPRAN, Surface Ocean PRocesses in the ANthropocene, http://sopran.pangaea.de/) in December 2012 was dedicated to study biogeochemistry and emissions of various climate-relevant atmospheric trace gases. It yielded several observational parameters that serve this study's purpose to explore the magnitude, causes, and impacts of near-surface $N_2O$ concentration gradients. The data set is complemented with near-surface hydrographic time series from a campaign using several ocean gliders during the subsequent cruises Meteor M92

and M93, carried out as part of the German collaborative research center SFB754, www.sfb754.de (Dengler and Krahmann, 2017a, b; Kanzow and Krahmann, 2017a, b, c, d, e, 2018). For cruise reports see Bange et al. (2013), Sommer et al. (2014), Lavik et al. (2013). On most of the ship stations during the December 2012 cruise, simultaneous profiles of conductivity-temperature-depth-oxygen ($CTD-O_2$, (Krahmann and Bange, 2016)) and discrete samples of $N_2O$ (Kock and Bange, 2016)



were collected (Fig. 1). These data were used to estimate the near-surface vertical $N_2O$ gradient, the stratification between 10 m and 5 m, the thickness of the top layer (cf. subsection 2.2.3) as well as the depth of the OMZ upper boundary - here defined by a $20\,\mu\,mol\,kg^{-1}$ oxygen threshold. The latter served to approximately locate the periphery to anoxic conditions, with a sharp oxygen gradient and with expected strong local $N_2O$ production, and will be called 'oxygen interface' in the

following. Four vertically high-resolution $N_2O$ profiles of the top 10 m were measured from a drifting Zodiac positioned at least 0.5 nm away from the research vessel (Fig. 1). The Zodiac sampling aimed at identifying near-surface $N_2O$ gradients not affected by ship-caused turbulence. The top 1 m was sampled by a submersible centrifugal pump with radial intake, providing water at a rate of about $0.5\,L\,min^{-1}$. For the water column from 1 m to 10 m a manually triggered 5 L-Niskin bottle was used, accompanied by a MicroCat to record pressure, temperature and salinity.

Underway $N_2O$ concentrations at 5.5 m were measured continuously from the ship's moon pool, and are used in this study to complement the Zodiac high-resolution $N_2O$ profiles. In order to estimate $N_2O$ 5.5 m-concentrations on station, only values obtained near the station were considered when the vessel was steaming, to avoid disturbances of the water column by the ship's maneuvering and dynamic positioning. Underway water temperature at the thermosalinograph intake at the ship's hull (at 3 m depth) together with the vertical displacement of the intake was used to create an along-track time series of estimated

near-surface stratification, in order to explore the association of strong near-surface stratification events and $N_2O$ gradients. Further, a campaign with 7 gliders in January and February 2013 (Thomsen et al., 2016), provided undisturbed near-surface hydrographic data with high temporal coverage for 4 local areas (Fig. 1). For these areas which are characterized by different wind conditions and different distances to land, 1-hour-resolution time series of stratification in the top 12 m could be composed. These time series served to estimate the occurrence and characteristics of multi-day near-surface stratification, and to

force a simple one-dimensional gas-transfer model of the top 12 m of the water column, aimed at producing time series of $N_2O$ distribution and outgassing for different stratifications and wind conditions.

## 2.2 Sample and data processing

### 2.2.1 $N_2O$ concentrations

For the discrete $N_2O$ measurements, 20-mL water samples were taken (three replicates per depth during $CTD-O_2$ casts,

six replicates per depth during high-resolution profiles). Following Kock et al. (2016), the samples were analyzed onboard by gas chromatography with electron capture detector (GC-ECD) after bringing a helium headspace to static equilibrium. The measurement uncertainty was estimated for each profile separately, from the distribution of residuals around the average profile, and lay typically in the range of 0.5 to 1 $nmol\,kg^{-1}$ (95% level) for the high-resolution profiles and in the range of 0.5 to 4 $nmol\,kg^{-1}$ (95% level) for the CTD profiles. $N_2O$ was also measured from a continuous seawater supply (pumped from

5.5 m depth) with a cavity enhanced absorption spectrometer coupled to a seawater/gas equilibrator (Arévalo et al., 2013). The response time of the equilibrator was 2.5 minutes (translating to a space scale of 750 m at a ship speed of 10 knots). The accuracy of 3-minute averages is $<0.5$ $nmol\,kg^{-1}$. A possible instrument drift, which is typically lower than 1 % per week, was corrected by a 6-hourly calibration of the measurement system (Arévalo et al., 2013).

### 2.2.2  CTD−O₂

Salinity , temperature, and oxygen profiles were obtained from a lowered SeaBird911plus CTD with dual conductivity and temperature sensors, plus added membrane-type oxygen sensors. Salinity was calibrated against water samples analyzed with a Guildline AutoSal salinometer. Oxygen was calibrated against water samples using a Winkler titration stand. No further

calibration of temperature sensors was performed. Accuracies are 0.002 K in temperature, 0.002 in salinity, $1\,\mathrm{\mu mol\,kg^{-1}}$ in oxygen for concentrations $\geq 5\,\mathrm{\mu mol\,kg^{-1}}$. We also use temperature profiles derived from a microstructure probe which was equipped with a Pt100 temperature sensor and a thermistor. The gliders carried unpumped CTDs that required a special treatment. Following Thomsen et al. (2016), the flow through their conductivity cells was derived from a glider flight model, a thermal lag hysteresis correction was applied, and derived temperature and salinity values were further calibrated against

shipboard CTD data from stations close to the glider position. Accuracy (rms) is 0.01 K in temperature and 0.01 in salinity.

### 2.2.3  Thickness of the top layer

We will use the term 'top layer' (TL) to refer to that layer which ranges from the ocean surface down to a layer of strong stratification, and whose interior is characterized by a relatively weak stratification or even homogeneity. In extreme cases when strong stratification extends to the surface, a TL will not exist. To coin a new term instead of using 'mixed layer' or

'mixing layer' is to avoid misunderstandings and misconceptions, as the varieties of definitions and criteria for the latter terms are ample, and sometimes the TL might rather match the mixed layer, sometimes the TL might better match a temporal mixing layer within the mixed layer. It is intended to have the top layer describe the layer of trapped water, and its thickness or 'top layer depth' (TLD) to describe the depth below which turbulent mixing is suppressed. Therefore we define the TLD based on a criterion relevant for the trapping process. The TLD is at the transition from the TL to the layer of suppressed mixing,

and matches the 'trapping depth' of Price et al. (1986), Fairall et al. (1996a), and Prytherch et al. (2013), who considered surface trapping by the diurnal warm layer cycle. Reported criteria are based on the argument that the trapping depth is set by self-regulation between the competing effects of stratification and shear instability and comes to sit where the gradient Richardson number (Ri) is about critical (Price et al., 1986; Fairall et al., 1996a; Prytherch et al., 2013; Soloviev and Lukas, 2014). Reported Ri criteria are 0.25 and 0.65, typical shear at trapping depth is 0.5 to $2 \cdot 10^{-2}\,\mathrm{s^{-1}}$ (Prytherch et al., 2013) or

$1 \cdot 10^{-2}\,\mathrm{s^{-1}}$ (Wenegrat and McPhaden, 2015), both derived from observations of diurnal warm layers. These values correspond to an $N^2$ range of $10^{-5}\,\mathrm{s^{-2}}$ to $10^{-4}\,\mathrm{s^{-2}}$ and match the $N^2$ range at trapping depth observed by Wenegrat and McPhaden (2015). We define TLD as the minimum depth where $N^2 \geq 10^{-4}\,\mathrm{s^{-2}}$, in order not to underestimate the trapping depth, and not to overestimate the resulting effects. This way to calculate TLD requires reliable density profiles up to the surface, which is given for the glider hydrographic surveys during January/February 2013. In contrast, the shipboard CTD profiles taken in

December 2012 are much less reliable in the top 10 m, because the ship's engines and maneuvring before and during CTD stations causes overturns and turbulence. This is also the reason why shipboard CTD data usually do not show near-surface density gradients of that strength we found in the glider data. For the lack of reliable density data we use for the ship CTD data an auxiliary but more robust criterion. It is based on temperature difference to the surface, and originally intended for mixed




layer detection, cf. Schlundt et al. (2014). The temperature profiles from the shipboard CTD were complemented by collocated temperature profiles from the microstructure probe to reduce uncertainty. To reduce the effect of ship-induced turbulence and under the assumption that any unstable stratification is artificially generated, the measured temperatures of the top 10 meters were sorted with highest temperatures at the surface. The depth criterion applied is a density increase compared to the surface

which is equivalent to a temperature decrease of $0.5°$ C while salinity is kept constant (Schlundt et al., 2014). This alternative top layer thickness estimate will be referred to just as surface layer depth, to illustrate that it is methodically different from TLD.

### 2.2.4 Underway estimate of stratification at 3 m depth

We used the water temperature measured at the thermosalinograph inlet near the ship's bow at nominal 3 m depth, and the

vertical movement of the inlet position relative to the water column, in order to derive estimates of the stratification at about 3 m depth while the ship was cruising. This was inspired by the strategy of scanning the near-surface range with bow mounted sensors by Soloviev and Lukas (1997). As the actual wave height and phase time series are unknown, the inlet position is calculated relative to the mean sea level, defined as average water level relative to the ship in immediate neighborhood of the ship. The vertical distance of the inlet relative to the mean sea level was estimated by rotating the vector of distance of the inlet

relative to the ship's centre of mass - first rotating around the ship's pitch axis, then around the ship's roll axis, resulting in

$$d_{inlet/sealevel} \approx -x_{inlet/com} \cdot sin\pi + (y_{inlet/com} \cdot sin\rho - z_{inlet/com} \cdot cos\rho) \cdot cos\pi + d_{com/sealevel} \qquad (4)$$

with $(x, y, z)_{inlet/com}$ as inlet position relative to centre of mass in ship coordinates, $x$ positive to bow, $y$ positive to starboard, $z$ positive up, $\rho$ roll angle positive for starboard down, $\pi$ pitch angle positive for bow up, $d_{com/sealevel}$ distance of centre of mass to sea level. Heave is not part of the transformation because it is assumed that the ship's centre of mass does only

negligibly move relative to the mean sea level. The transformation is further only approximate because vertical displacement of the water column at 3 m from wave orbitals or a possible correlation of $d_{inlet/sealevel}$ and actual sea level at the inlet position could not be taken into account. As the time series of recorded data of temperature and vertical position are not reliably synchronous, the vertical temperature gradient is estimated by the square root of the temperature variance divided by the square root of the vertical distance variance. The used variances are variances of residuals relative to a 200-second low-

pass. The entire procedure assumes that the temperature variance is dominated by the vertical temperature gradient. However, horizontal temperature variability on short scales, vertical movements of the water column, and sensor noise add to temperature variance. The lower limit of the calculated $N^2$ of about $10^{-5}\,s^{-2}$, which we find in the cruise data (cf. Fig. 4), is probably caused by this additional variance. The salinity required to convert the temperature gradient into stratification is taken from the thermosalinograph record, using the average salinity during the respective time bin, i.e. assuming a vertical salinity gradient of

zero. The derived $N^2$ time series is not used quantitatively due to the described limitations, but allows qualitatively identifying spatiotemporal variations in near-surface stratification.





### 2.2.5 Wind speed at 10 m and cloud radiation

Wind speed at 10 m height was needed to estimate gas exchange fluxes. 10 m wind speed during the ship cruise was derived by converting the wind speed measured at 34 m height at the ship using the COARE algorithm for non-neutral atmospheric conditions (Fairall et al., 1996b). 10 m-wind is the wind speed that exerts the same wind stress on the water surface as the measured 34 m-wind, under the measured atmospheric conditions. In order to account for the integrated effect of the varying wind in the gas exchange estimates, wind speed was rms averaged using a cutoff radius in time and space of 6 h and 5 nm, respectively, around the time and position of $N_2O$ sampling. The averaging scales had been chosen after inspecting the underway $N_2O$ dataset for typical spatial scales of variability during cruising and for typical scales of temporal variability at station. Averaging was quadratic in order to estimate an effective wind speed that induces the same transfer velocity as the integrated time series of varying transfer velocities, acknowledging that transfer velocities can be well described as proportional to wind speed squared in the lower to medium wind speed range (Garbe et al., 2014), a range that was encountered during most of the cruise. For the glider time series we used (1) daily wind fields from Metop/ASCAT scatterometer retrievals (http://cersat.ifremer.fr, Bentamy and Croize-Fillon (2012)) that were interpolated to the positions of the gliders, and (2) wind speed from collocated ship records (distance $< 0.3°$) that was allocated to parts of the glider hydrographic time series, i.e. only when the ship was nearby. For the latter positions, also the long wave radiation (LWR) attributable to cloud cover was calculated, from incoming LWR minus clear sky LWR. These ship based observations of wind and cloud-caused LWR will serve to investigate conditions for multi-day near-surface stratification, but due to the gaps in the data cannot serve to force the $N_2O$ gas-transfer model of subsection 2.2.7.

### 2.2.6 $N_2O$ flux densities by air-sea gas exchange, and relative flux error

In order to estimate the $N_2O$ flux density ($\mathrm{nmol\,m^{-2}\,s^{-1}}$) from or to the ocean, the bulk flux parameterization of Nightingale et al. (2000) was used with a Schmidt number exponent of $n = -0.5$. The transfer velocity here only depends on wind speed with a quadratic law, and is of medium range within the multitude of transfer velocity parameterizations (Garbe et al., 2014). We also calculate a relative flux error (similar to Soloviev et al. (2002)) which quantifies the bias if not calculating the flux density based on the proper bulk concentration but based on a differing concentration somewhere in the near-surface:

$$R = \frac{\Phi_{ns} - \Phi_{bulk}}{\Phi_{bulk}} = \frac{\Phi_{ns}}{\Phi_{bulk}} - 1 = \frac{c_{ns} - c_{eq}}{c_{bulk} - c_{eq}} - 1 \tag{5}$$

with $\Phi_{bulk}$ the flux density based on bulk concentration $c_{bulk}$, $\Phi_{ns}$ the flux density based on concentration $c_{ns}$, and $c_{eq}$ the concentration in equilibrium with the atmosphere. $c_{eq}$ was calculated following Weiss and Price (1980), using an $N_2O$ mole fraction in dry air of 325 ppb. $R$ can be interpreted as the overestimation percentage of the gas exchange rate if the estimate is based on a concentration $c_{ns}$. The advantage of this relative measure of bias is that it shows the impact of the Delta c sampling issue in a clear way independent of the actual value of the transfer velocity and its issues, and abstracting from the actual concentration level of the local $N_2O$ profile. Certainly, transfer velocities and $N_2O$ concentrations will have to be taken into account when estimating the integrated effect of near-surface stratification on regional emission rates.

### 2.2.7  One-dimensional gas-transfer model of the surface trapping mechanism

It is to be investigated if the observed vertical near-surface $N_2O$ gradients can be caused by near-surface stratification alone. Further, we want to compare the impact of multi-day near-surface stratification versus the impact of just diurnal episodes of near-surface stratification. For these purposes, a model is used which simulates the surface trapping mechanism in a straight-

forward and simplified manner by vertical one-dimensional transport processes. The model represents the top 12 m of the water column, and takes into account $N_2O$ supply from below, air-sea gas exchange at the surface, and the suppressed mixing that is caused by a thin near-surface stratified layer. That thin stratified layer is simplified to be an interface of complete mixing inhibition, which divides the water column into two separate layers. The two layers (top layer/lower layer) are idealized to be each immediately and completely mixed. The interface of complete mixing inhibition represents the TLD and can shift up and

down in the water column, independent of water movements. That means that top and lower layer can change thicknesses, and entrain water of each other, which leads to the exchange of $N_2O$ between the layers. The model is constrained by observational data from 4 locations in the upwelling regime (region I, II, III, IV in Fig. 1). The locations represent different grades of near-surface stratification, from domination by diurnal episodes to domination by multi-day events. The corresponding 4 time series of TLD stem from glider hydrographic near-surface profiles in January/February 2013 (cf. subsections 2.1, 2.2.2,

and Thomsen et al. (2016)). Density time series of hourly resolution in the top 12 m were assembled from shorter time series of different gliders that were passing through regions I to IV. The density time series were then low-pass-filtered (12-hour half power, 3-hour cut off) to remove density changes that are only caused by vertical movements of the water column due to internal waves and would otherwise cause spurious exchange between the two layers. TLD was determined as the shallowest depth where stratification was stronger than $N^2 = 10^{-4}\,s^{-2}$ (see subsection 2.2.3). Air-sea gas exchange was calculated via

the Nightingale et al. (2000) parameterization from the actual simulated $N_2O$ concentration of the top layer, from $c_{eq}$ based on surface temperature and salinity of the glider hydrographic data, and from transfer velocity calculated from wind speed (see subsection 2.2.5). $N_2O$ supply from below was determined based on the assumptions that observed $N_2O$ concentrations at 20 m depth can be treated as steady-state, thus are understood as constant boundary values, and that $N_2O$ transport into the lower layer is by turbulent mixing. Actual 20 m-concentrations were taken from discrete $N_2O$ profiles of December 2012 that were

both nearby to region I to IV and situated at land distances that corresponded to those of region I to IV. Chosen values were 50, 30, 40, 60 $nmol\,kg^{-1}$, respectively. The supply flux density was then calculated as $\Phi = \rho \cdot K \cdot \nabla N_2O$ with $\rho$ water density, $K$ vertical exchange coefficient, and $\nabla N_2O$ vertical gradient of $N_2O$ concentration. The $N_2O$ gradient is the difference between 20 m-concentration and the concentration in the lower layer, divided by the distance between 20 m and the temporary centre depth of the lower layer. In order to get an estimate of the range of the vertical exchange coefficient $K$, $K$ was determined from

microstructure measurements at stations where strong shallow stratification between two weakly stratified layers was clearly present. There, vertically averaged $K$ was determined for the depth range from below the TLD down to 20 m. For details of K estimation from velocity microstructure see Fischer et al. (2013). The observed $K$ values ranged from $10^{-5}\,m^2\,s^{-1}$ to near $10^{-2}\,m^2\,s^{-1}$ with median $10^{-4}\,m^2\,s^{-1}$ and mean $10^{-3}\,m^2\,s^{-1}$. After having chosen a value for $K$ and which region I to IV to be simulated, the model is forced by cyclic application of according wind and TLD time series until cyclic equilibrium. In



result, the model produces time series of $N_2O$ concentration vs. depth, so that time series of measurement bias R vs. depth can be obtained and compared to observations.

## 3 Results

The four off-ship high-resolution $N_2O$ profiles (A to D) which are not affected by ship-caused stirring show that near-surface

$N_2O$ gradients do generally exist in the Peruvian upwelling region (Fig. 2). The $N_2O$ gradients are of different strength but all downward or zero, and go in hand with thin homogeneous top layers of 1 to 5 meters thickness. They strengthen with decreasing land distance and lower wind speed. And they are very similar in shape to the corresponding density profiles, i.e. a stronger $N_2O$ gradient is also associated with stronger stratification.

Discrete $N_2O$ samples from the closest shipboard CTD profiles are consistent with the off-ship profiles, despite some dis-

tance in space and time. Underway $N_2O$ data during station approach/leaving are from distinctly larger distance in space and time than the discrete $N_2O$ samples and vary stronger, though still match the general pattern. Particularly at site C the underway data span the entire concentration range of the top 10 m. The consistency of off-ship, discrete, and mean underway $N_2O$ concentrations suggests larger regions of at least some miles extent to be basically horizontally homogeneous in the top 10 m, while the variability of underway $N_2O$ concentrations particularly at site C suggests that vertical motions (most likely due to

internal waves) are superposed transferring water from different nominal depths to the sample inlet at 5.5 m. Such variability is not visible in the discrete $N_2O$ samples of profile C, because these were projected onto the mean density profile which was observed during the off-ship sampling. I.e. profile C does explicitly not show variability caused by internal wave motion, which was strong in the top meters at that site.

In order to further explore the spatial distribution and the conditions that lead to near-surface $N_2O$ gradients, the data set is

complemented by the topmost ship-based $N_2O$ samples collected during December 2012. By taking into account these data we accept the enhanced uncertainty in allocating $N_2O$ concentrations to depths which arises from ship induced disturbances in the top 10 m of the water column. On the other hand we have shown a consistent behavior of off-ship and shipboard $N_2O$ samples at sites A to D. The ship-based data allow to examine the $N_2O$ difference between about 5 m and 10 m depth. This provides a dataset of 45 near-surface $N_2O$ gradient estimates, as plotted in Fig. 3a as function of distance to land. The encountered

$N_2O$ gradients are mostly downward, i.e. negative with the convention of the z-axis pointing upward, but occasional upward gradients occur very close to the coast. Far off coast, gradients are mostly insignificant. The compilation shows that stronger $N_2O$ gradients exist than observed at the off-ship high-resolution stations, and suggests a zoning into neutral ('no') gradients off 60 nm, downward gradients between 60 nm and 6 nm, and upward gradients inland of 6 nm. These zone limits are peculiar for the sampling depth between 5 m and 10 m, and would probably take different values for gradients at other sampling depths.

Note that the profiles' behavior shallower than 5 m is unknown here, so we cannot exclude that profiles of upward gradient between 5 m and 10 m still exhibit a downward gradient in the top meters. Note as well that the high-resolution profiles tended to not exhibit their strongest gradients between 10 m and 5 m, suggesting that other profiles are likewise and thus stronger gradients than shown in Fig. 3a might exist. The single occurrence of a strong $N_2O$ gradient at 70 nm off shore coincides

with a shallower mixed layer and less oxygen below the mixed layer than expected at that open ocean position. The sea surface temperature field at the time of sampling shows a filament reaching from the coast to the station position. Those aspects suggest that coastal water already carrying a downward $N_2O$ gradient has been transported to the open ocean.

Fig. 3b shows that strong $N_2O$ gradients (downward and upward) are confined to strong stratification, with a threshold
buoyancy frequency of about $N^2 = 10^{-4}\,\mathrm{s}^{-2}$. Following the arguments in subsection 2.2.3 that during surface trapping the trapped top layer is isolated from waters below by already somewhat weaker stratifications of $N^2$ between $10^{-5}\,\mathrm{s}^{-2}$ and $10^{-4}\,\mathrm{s}^{-2}$, this indicates that the strong $N_2O$ gradients are associated with surface trapping.

How much time would be needed to form the observed $N_2O$ gradients by surface trapping and air-sea gas exchange? The shipboard discrete $N_2O$ data allow a rough estimate for the majority of profiles with significant gradients, namely the downward
ones, with 5 m-concentration < 10 m-concentration (Fig. 3c). The calculation assumes that initially homogeneous $N_2O$ profiles at 10 m-concentration got trapped from 5 m depth up to the surface, while no horizontal $N_2O$ transport and no $N_2O$ supply from below occurs. Then the top 5 m are depleted by air-sea gas exchange, until they reach the observed 5 m-concentration. Thus the difference between 10 m-concentration and 5 m-concentration is the supposed $N_2O$-deficit that arose during past hours of isolation of the top 5 m, under assumed constant wind conditions as observed during sampling. Taking into account
that we expect the top 5 m to exhibit a downward or neutral gradient (cf. Fig. 2), the $N_2O$ deficit calculated in this simplified way is actually expected to be a lower bound to the real amount of $N_2O$ that has been emitted. Together with the assumption of no $N_2O$ supply from below this means that the calculated time spans rather underestimate the necessary duration of surface trapping. The strongest quarter of $N_2O$ gradients in Fig. 3c needs isolation periods of distinctly more than 24 hours, i.e. multi-day near-surface stratification, and there is some other strong gradients with isolation periods shorter than 24h, that however
still comprise the entire previous night. Profiles of upward gradient between 10 m and 5 m will be discussed in subsection 4.3.

The suggestion that multi-day near-surface stratification exists and is not rare, and that it is associated with the strongest near-surface $N_2O$ gradients, is further supported by additional observations. Fig. 4 aligns the shipborne along-track time series of estimated $N^2$ at 3 m depth during December 2012 with the observed $N_2O$ gradients. The time series of 3 m-stratification shows a distinct diurnal cycle with maximum stratification around 15:00 local time. We aimed to subtract that diurnal cycle of
near-surface stratification, in order to mimic a time series of the local nighttime $N^2$ minimum, and in this way detect locations where near-surface stratification probably survived the previous night and can be called multi-day near-surface stratification. Interestingly, the diurnal cycle is much better removed in logarithmic space than in linear space; so we calculated a mean diurnal cycle of $log_{10} N^2$, scaled it with an offset such that the minimum of $(log_{10} N^2 + offset)$ equals zero, then subtracted this scaled mean diurnal cycle from the time series of $log_{10} N^2$. The nonlinearity of the diurnal evolution of near-surface
stratification might be due to the fact that pre-existing stratification will suppress turbulent mixing and increasingly promote surface trapping of heat during daytime, thus self-perpetuate the increase of near-surface stratification. Fig. 4 shows that the strongest $N_2O$ gradients come in 3 clusters (i.e. around day 5, 10, and 15, respectively), and they are associated with minimum nighttime stratification of order $N^2 = 10^{-4}\,\mathrm{s}^{-2}$, which is strong enough to assume surface trapping (subsection 2.2.3). The clusters suggest the existence of larger regions of multi-day near-surface stratification that have been cut through by the cruise
track. Direct observational evidence for multi-day near-surface stratification in the form of stratification time series in fixed





regions comes from 4 local hydrographic time series obtained during the glider campaign in January/February 2013 (Fig. 5). The time series in regions I to IV (see Fig. 1) show different grades of persistence of near-surface stratification, ranging from a classic diurnal warm layer periodicity with regular nighttime mixing (I) to a strong stratification layer not retreating deeper than 2 m from the surface for several days in a row (IV). Conditions that promote the occurrence of multi-day near-surface

stratification were examined for the glider data at nights when glider positions and ship positions were collocated (distance $\leq 0.3°$ in latitude/longitude), so that wind speed and long wave radiation from clouds could be assigned to thicknesses of the homogeneous top layer (Fig. 6). The data show that at low to moderate wind (0 to 6 m s$^{-1}$) it is possible to find near-surface stratification persisting all night, the main prerequisite of multi-day near-surface stratification. Below wind speeds of 3 to 4 m s$^{-1}$ multi-day near-surface stratification even seems certain. Additional cloud cover supports the persistence of near-surface

stratification. Unfortunately the glider time series could not be accompanied by N$_2$O measurements, so that a co-occurrence of the glider-observed periods of multi-day near-surface stratification with a progressing formation of strong N$_2$O gradients can only be checked for plausibility. This check is done with the 1-D gas-transfer model introduced in subsection 2.2.7, simulating within its simple setup the surface trapping mechanism and the formation of N$_2$O gradients. The model is forced with the glider time series of TLD and with ASCAT daily wind. Fig. 7 shows N$_2$O distributions as function of depth which result from

the model runs with applied forcings of region I to IV, displayed as distributions of relative flux error $R$ or flux overestimation (subsection 2.2.6). $R$ is insensitive to the actual N$_2$O supply from below, both for the range of assumed 20-m concentrations and for the range of vertical turbulent diffusivity from $10^{-5}$ m$^2$ s$^{-1}$ to $10^{-2}$ m$^2$ s$^{-1}$. This insensitivity is plausible, because $R$ can be expressed as $\frac{c_{ns}-c_{bulk}}{c_{bulk}-c_{eq}}$, $(c_{ns}-c_{bulk})$ is proportional to the N$_2$O flux from the lower layer (with $c_{ns}$) to the top layer (with $c_{bulk}$), $(c_{bulk}-c_{eq})$ is proportional to the N$_2$O flux from the top layer to the atmosphere, and in the model equilibrium both

fluxes are equal on average. This way, expressed as $R$, modeled N$_2$O gradients can be advantageously compared to observed gradients without considering the magnitude of supply flux. It is just the impact of surface trapping on gradient formation that is compared between model and observed N$_2$O profiles. The results in Fig. 7 show that the model produces distributions of $R$ that comprise the observed $R$ of the high-resolution N$_2$O profiles. I.e. the observed N$_2$O gradients during December 2012 are within the range that was modeled in accordance with observed surface trapping scenarios. An increase in the number of

multi-day events in the TLD time series I to IV leads to increasingly higher $R$ values, i.e. increasingly stronger N$_2$O gradients are expected on average.

## 4   Discussion

### 4.1   The role of multi-day near-surface stratification for near-surface gas gradients

We will argue here that multi-day persistence of near-surface stratification is able to explain the formation of strong near-surface

gas gradients, and furthermore that it is unlikely to achieve strong gas gradients through near-surface stratification on shorter timescales. The basic linkage of near-surface stratification and vertical gradients of any property in the near-surface ocean has been established (particularly plainly stated by Soloviev and Lukas (2014)), and is attributed to turbulence suppression in the temporally stratified layer, i.e. to surface trapping. However studies dealing with consequences of near-surface stratification



generally focus on short timescales, usually on the diurnal warm layer cycle (Soloviev et al., 2002; Kawai and Wada, 2007; Gentemann et al., 2009; Wenegrat and McPhaden, 2015). Prytherch et al. (2013) mention the possibility of pre-existing stratification at sunrise (i.e. incomplete erosion of stratification during the night and longer timescales of near-surface stratification are implied), and observe subsequent amplification of surface warming, but they do not explore further consequences. Our

database and results allow to extend the view to the multi-day timescale. In this respect our results show firstly, that multi-day near-surface stratification is not rare, lasts up to several nights in a row, and that remaining stratification at sunrise is strong of order $N^2 = 10^{-4}\,\mathrm{s}^{-2}$ and more (Fig. 5). Conditions which support the endurance of stratification through the night and thus multi-day timescales are basically the same that promote near-surface stratification on shorter timescales, that is low wind energy input and low heat loss (Fig. 6). Secondly, observations show that the absolute near-surface $N_2O$ gradient is positively

related to the strength of near-surface stratification (Fig. 2, Fig. 3b), such that the observation that multi-day stratification is strong results in the expectation of associated strong $N_2O$ gradients. Thirdly, the duration of near-surface stratification can be directly related to the strength of near-surface $N_2O$ gradients. This is indicated by three lines of observations and analyses. (i) During the cruise in December 2012, clusters of multi-day stratification coincided with clusters of strongest $N_2O$ gradients (Fig. 4). (ii) When estimating necessary trapping times to produce observed $N_2O$ gradients (Fig. 3c), the strongest quarter of

gradients can only be caused by multi-day trapping. (iii) When on the other hand estimating $N_2O$ gradients caused by observed trapping conditions (process model with observed TLD time series, Fig. 7), strong gradients become more and more likely with more frequent occurrence of multi-day events.

Until here, the line of evidence supports that multi-day near-surface stratification can explain strong near-surface gradients. To go beyond this, Fig. 3c and Fig. 7, and also the results of Soloviev et al. (2002) suggest that substantial gas gradients are

not only made possible by, but even need trapping times beyond the typical up to 12 hours of the diurnal warm layer cycle. 'Substantial' is unfortunately vague here, because the strength of gradients cannot be directly compared between the figures. Fig. 7 indicates that region I which is dominated by the diurnal cycle is good for a typical $R$ of 10 %, while region IV which is dominated by multi-day near-surface stratification exhibits $R$ of 50 % to 100 %. The transition between diurnal and multi-day domination may be seen in regions II and III with $R$ about 30 %. This is in line with Soloviev et al. (2002) who find a

maximum $R$ of 30 % in their investigation of gas gradients caused by the diurnal warm layer cycle. For the gradients of Fig. 3c, information on concentrations above 5 m depth is lacking, so $R$ cannot be calculated. However we can still roughly estimate $R$ by using the concentration at 5 m for $c_{bulk}$, and using the concentration at 10 m for $c_{ns}$, as is done in Fig. 8. This results in a threshold for $R$ of 30 % to 50 %, above which gradients can only be achieved by multi-day near-surface trapping. Overall, these three independent estimates indicate that near-surface stratification at diurnal timescale can only account for gradients

worth $R = 30$ % or less.

Can we understand this better, that mainly the trapping time seems to play such an important role for gradients? Other factors as TLD and wind speed are involved in the effectiveness of the surface trapping mechanism, but it seems they only occur in combinations which lead to necessary trapping times on multi-day scale in order to cause substantial $N_2O$ gradients. To gain some insight, we examine the formation of downward $N_2O$ gradients in a very simplified setting, and work out the time and

TLD dependence of relative emission bias $R$ (as a measure for gradient strength). Assumed is an initially homogeneous water



column of concentration $c_0$ which becomes stratified at the depth TLD at time $t_0 = 0$. The stratification immediately causes a complete shutdown of $N_2O$ supply from below, such that only gas exchange with the atmosphere acts and diminishes the concentration $c_{TL}$ in the TL. In the following we will call this simplified process model the 'shutdown model'. The difference to the 1-D gas-transfer model of subsection 2.2.7 is the lack of vertical movement of the TLD which would permit $N_2O$ supply

from below through entrainment. Using a bulk parameterization, the outgassing flux density will be $\Phi = k_w \cdot (c_{TL} - c_{eq})$, and the change in top layer concentration with time $\frac{dc_{TL}}{dt} = -\frac{\Phi}{TLD} = -\frac{k_w}{TLD} \cdot (c_{TL} - c_{eq})$. The solution is $c_{TL} = c_{eq} + (c_0 - c_{eq}) \cdot exp(-\frac{k_w}{TLD} \cdot t)$, such that

$$R = \frac{c_0 - c_{eq}}{c_{TL} - c_{eq}} - 1 = exp(\frac{k_w}{TLD} \cdot t) - 1. \tag{6}$$

The decisive timescale here is $\frac{TLD}{k_w}$ and the necessary trapping time to reach a certain $R$ is

$$T_{trap} = \frac{TLD}{k_w} \cdot log(R + 1). \tag{7}$$

For $k_w$ we choose the transfer velocity of Nightingale et al. (2000) which after scaling to the $N_2O$ Schmidt number is a function of wind speed $u_{10}$ only, $k_w = (\frac{2}{9} \cdot u_{10}^2 + \frac{1}{3} \cdot u_{10}) \cdot (\frac{Sc_{N2O}}{600})^{-0.5}$. To estimate trapping times $T_{trap}$ as a function of $R$ and TLD, we use TLD from glider observations, and corresponding $u_{10}$ from nearby ship time series, which were already employed to investigate the conditions for multi-day stratification (Fig. 6). Displaying $R$ as function of $T_{trap}$ and TLD (Fig. 9

left panel) shows that TLD has an effect, but $R$ proves to be more sensitive to changes in $T_{trap}$ than in TLD, within the observed range of values. This can be explained by the relation of TLD and $k_w$ (or $u_{10}$): weaker wind which tends to accompany thinner TL leads to a reduction in gas exchange so that gradient formation is only weakly intensified with decreasing TLD. However, for very thin TL with TLD $\leq 0.5$ m, trapping on diurnal timescale might produce $R > 30\%$. Unfortunately, this is outside of our observational evidence.

So far we evaluated the strength of gas gradients in terms of relative flux overestimation $R$. If we want to evaluate the absolute impact of gas gradients on gas flux estimates, the transfer velocity and the actual gas concentration have to be accounted for as well. Keeping the shutdown model that was introduced just above, and defining the absolute flux bias $\Delta\Phi$ as difference between the flux estimate based on concentration $c_0$ and the flux estimate based on concentration $c_{TL}$, we get

$$\Delta\Phi = k_w \cdot (c_0 - c_{eq}) - k_w \cdot (c_{TL} - c_{eq}) = k_w \cdot (c_0 - c_{TL}), \tag{8}$$

and using the definition of $R$ (equation 5),

$$\Delta\Phi = k_w \cdot R \cdot (c_{TL} - c_{eq}) = k_w \cdot \frac{R}{R+1} \cdot (c_0 - c_{eq}). \tag{9}$$

As there is no data for $c_0$ to accompany the relation between $k_w$ and TLD, $\Delta\Phi$ itself cannot be calculated, but we will examine the term

$$\frac{\Delta\Phi}{c_0 - c_{eq}} = k_w \cdot \frac{R}{R+1} = B \tag{10}$$



which can be interpreted as a specific absolute flux bias per unit supersaturation. Comparing $B$ for different conditions means to assume that $c_0$ is independent of the conditions, while TLD and $c_{TL}$ react to wind speed and trapping time. Fig. 9 (right panel) shows that $B$ is practically independent of TLD. This means, the enhancing effect on $B$ of a stronger gas gradient which comes with a thinner TL, is fully compensated by the diminishing effect on $B$ of the lower total gas transfer due to the lower

wind speed which enabled the thinner TL in the first place.

Thus we may conclude from this subsection that (i) the trapping time is decisive for the formation of gas gradients of high impact on gas exchange estimates (Fig. 9), and building on this, (ii) multi-day near-surface stratification can explain the observed gas gradients (Figs. 5 and 7), while (iii) substantial flux bias is not to be expected from near-surface stratification at diurnal or shorter timescale (Figs. 7, 8, and 9).

## 4.2 Moderate wind speed causes strongest gas exchange bias

Using the shutdown model of subsection 4.1 a bit further, the timescale $\frac{TLD}{k_w}$ as a function of wind speed $u_{10}$ (Fig. 10 left panel) suggests that there exists an optimum wind range for gas gradient formation. Gas gradients that cause a particular relative gas exchange bias $R$ are reached after a trapping time that is proportional to the timescale $\frac{TLD}{k_w}$ (cf. equation 7), and can thus be achieved in shortest time for moderate wind speeds between about 3 and 6 m s$^{-1}$. That means in this wind speed range it

should be most likely to observe strongest near-surface gradients. For wind below 3 m s$^{-1}$, gas exchange weakens while TLD remains about constant (cf. Fig. 6). For wind above 6 m s$^{-1}$, a more than proportional TLD increase outweighs the effect of increased gas exchange.

In order to examine the absolute gas exchange bias, Fig. 10 (right panel) shows the wind speed dependence of specific flux bias $B$, as introduced in subsection 4.1. $B$ depends on trapping time, but the functional shape of $B(u_{10})$ proves to be

independent of $T_{trap}$ (at least up to $T_{trap}$ = 48 hours), such that different $T_{trap}$ mainly cause a factor in $B$ or a constant offset in $log_{10} B$. We arbitrarily chose $T_{trap}$ = 12 hours to produce Fig. 10 (right panel). Again, the moderate wind range of 3 to 6 m s$^{-1}$ stands out. This time, for wind below 3 m s$^{-1}$, low $R$ and low air-sea gas exchange both mutually act to diminish flux bias. For wind above 6 m s$^{-1}$, $B$ is admittedly high, but practically the gas gradient is no longer a measurement issue, as TLD becomes greater than 5 to 10 m (cf. Fig. 6), and routine near-surface measurements now happen within the TL.

## 4.3 Spatial pattern of N$_2$O gradients in the Peruvian upwelling region

The previous insights lead us to propose an explanation for the observed distribution of near-surface N$_2$O gradients in the Peruvian upwelling region, particularly the qualitative zonation seen in Fig. 3a. There are several parameters in the upwelling region which are related to the distance to land (Fig. 11). Wind speed slows down toward the coast and sets favorable conditions for enhanced near-surface stratification and reduced top layer thickness near the coast. The favorable wind speed range for

gas gradient formation of 3 to 6 m s$^{-1}$ (subsection 4.2) is covered more and more frequently toward the coast. The oxygen interface is shoaling toward the coast, due to upwelling and more intense biological production, and subsequently more intense oxygen consumption at depth (Pennington et al., 2006). It reaches extremely shallow depths of about 10 m depth near coast, which however is not unusual (Hamersley et al., 2007; Gutiérrez et al., 2008). The oxygen interface is connected to peripheral





hotspot production of $N_2O$ (cf. introduction), thus we expect to find a shoaling strong local $N_2O$ source as well. Even if $N_2O$ production by nitrification is probably inhibited by light (Ward, 2008), we consider the local conditions favorable to sustain a shallow $N_2O$ source near the coast: denitrification and nighttime nitrification can intensely produce $N_2O$ in a near-surface oxygen interface that exists below the TLD for multi-day periods, and even during daytime we observed very high

chlorophyll content such that light absorption at 5 to 10 m depth may have been strong enough to allow for daytime nitrification. Fig. 11 shows that the depth of the shallowest local $N_2O$ maximum and the depth of the oxygen interface coincide, although with large variability superposed. This leads us to generally link the $N_2O$ maximum to the oxygen interface and peripheral hotspot $N_2O$ production, a conclusion also made by Ji et al. (2015) after investigating the metabolic activity of $N_2O$ producing microorganisms. This linkage is why we fit the shoaling of the oxygen interface and the shoaling of the $N_2O$ maximum by

the same line. Altogether the previous considerations lead to the following scheme of processes affecting the pattern of $N_2O$ concentration: (i) accumulation of $N_2O$ is favored below the TLD, because $N_2O$ is produced below the TLD and at the same time surface trapping slows down $N_2O$ loss toward the TL; (ii) $N_2O$ diminishes toward the surface, because in the TL it is reduced by gas exchange; (iii) $N_2O$ below the oxygen interface diminishes toward the deep due to an increasing influence of active $N_2O$ loss processes toward the anoxic part of the OMZ. The resulting principal shape of the $N_2O$ profile is characterized

by a local $N_2O$ maximum below the TLD at about the oxygen interface depth, and it shoals toward the coast because TLD and oxygen interface both shoal. Further the $N_2O$ maximum becomes more intense due to enhanced $N_2O$ production and more effective surface trapping toward the coast. A compilation of more and past $N_2O$ measurements off Peru (Kock et al., 2016) confirms this first order scheme.

Accepting this principal spatial structure, the horizontal zonation of observed $N_2O$ gradients (Fig. 3a) is immediately plausi-
ble as a consequence of scanning the tilted $N_2O$ field at a constant sampling depth. The two critical points are the land distance where the top layer depth becomes shallower than the sampling depth, and the land distance where even the oxygen interface becomes shallower than the sampling depth (Fig. 11). These critical points limit and define three zones, the offshore zone with no observed gradient when sampling above the top layer depth because $N_2O$ should be homogeneous within the TL, the near-coastal zone with downward gradient when sampling between top layer depth and oxygen interface/$N_2O$ maximum, and the

coastal zone with upward gradient when sampling below the oxygen interface/$N_2O$ maximum. Arguments are in the literature for a lower oxygen threshold of maximum $N_2O$ production than the $20\,\mu mol\,kg^{-1}$ we use, e.g. $< 10\,\mu mol\,kg^{-1}$ (Ji et al., 2015). Anyway, both 10 and $20\,\mu mol\,kg^{-1}$ oxygen isosurfaces are mostly positioned very close to the sharp oxycline - often beyond the practical uncertainty from which depth exactly the sampled water is from - and with standard CTD instrumentation and Winkler calibration, oxygen concentrations far away from $5\,\mu mol\,kg^{-1}$ are preferable for less uncertainty. So $20\,\mu mol\,kg^{-1}$ is

a practical choice to mark the approximate position of the oxygen interface.

The fraction of profiles in the coastal zone which show upward gradients at 5 to 10 m depth seems particularly interesting, because they are very high in $N_2O$ at 5 m and thus could be very important for the total $N_2O$ emission of an upwelling region. However, the behavior of $N_2O$ above 5 m is unknown. Likely is that a downward gradient from some point on up toward the surface will be present, because the occurrences of upward gradient profiles were at low wind conditions with very stable near-

surface stratification, so that long-duration surface trapping should be expected. The encountered wind speed of generally below



$3\,\mathrm{m\,s^{-1}}$ would though suggest that very long trapping times are necessary to produce strong downward gradients. In analogy to the process understanding of the downward gradient profiles farther offshore, the upward gradient profiles might be seen as an expression of local $N_2O$ production at the shallow oxygen interface. In this case a very strong and very shallow production is suggested to occur in a high productivity environment less than 5 m from the surface. However, while some upward gradient

profiles indeed show a coincidence of highest measured $N_2O$ concentration at the depth of the oxygen interface, others are highest still above the oxygen interface, at oxygen levels larger than $100\,\mathrm{\mu mol\,kg^{-1}}$. Kock et al. (2016) found that maximum $N_2O$ concentrations near the coast were indeed uncorrelated to the oxygen level. They discuss this to be an expression of strong time variability of oxygen conditions, i.e. the patchiness in the $N_2O$ distribution to be due to different oxygen histories, including some events of high $N_2O$ production at near anoxic level with resulting high $N_2O$ concentrations which are still

captured after mixing with water of higher oxygen level. This explanation would still leave surface trapping plus (transient) peripheral hotspot production as dominant processes in the near coast zone. However it can't be ruled out that other processes are involved as well.

## 4.4 Impact of near-surface $N_2O$ gradients on bias of total emission estimate

The impact of near-surface stratification on gas exchange seemed low so far, according to the rare studies. A study on oxygen

gradients and fluxes in the open ocean during the GasEx98 project (Soloviev et al., 2002) found weak gas gradients (average systematic oxygen flux overestimation of 4 % across the top 4 m, with peak maxima of 30 % in calm conditions). A study on oxygen and $CO_2$ near-surface gradients in different open ocean regions (Calleja et al., 2013) found large variability of upward and downward near-surface gas gradients in the top 8 m, which however was unsystematic with the mean gradient not significantly different from zero (their Fig. 2). However, the present study with its different conditions (upwelling region

instead of open ocean; tendency toward multi-day surface trapping; a gas which is basically biologically inactive in the near-surface) suggests a higher impact on gas exchange. We find stronger gas flux overestimations $R$ of median 12 %, mean 37 %, a 95 %-interval of [-40 % 180 %] and a maximum of 770 % across the depth range from 10 m to 5 m from ship based profiles, and the $N_2O$ gradients are systematically downward with exception of the coastal zone (Fig. 3a). As the observed near-surface $N_2O$ gradients are both strong and systematic, we expect a non-negligible bias on $N_2O$ emission estimates for the entire

region of the Peruvian upwelling. Assuming that the conclusions of the previous subsections are valid, that measurements are representative, and building on model results, we will estimate the total emission bias in the following, if relying on bulk flux parameterizations and sampling at 10 to 5 m depth.

For this purpose, stationwise $N_2O$ fluxes are calculated using the Nightingale bulk flux formulation, from 10 m-measurements, 5 m-measurements, and 'true' bulk concentrations, using collocated shipborne wind speeds (cf. subsection 2.2.5). The 'true'

bulk concentrations are the main issue here, and, apart from the measured values of the 4 high-resolution profiles, have to be estimated. For this purpose we take advantage of common features of profiles in the three zones (Fig. 3a, Fig. 11), and assume that near-surface gradients in each zone obey common distributions, which we estimate from the model results (Fig. 7) and the high-resolution profiles (Fig. 2). For the offshore zone we assume no multi-day stratification, as found in region I and in high-resolution profile B, and choose a normal distribution for $R$ with mean zero and standard deviation 0.1, i.e. N(0,10 %). For





the near-coastal zone we use regions II to IV and high-resolution profiles A, C, and D, which all are from the zone of downward gradients, and choose N(40 %,20 %) as $R$ for 10 m-concentrations and N(30 %,20 %) as $R$ for 5 m-concentrations. The coastal zone is particularly uncertain, as we have no observations for the behavior of the upward-gradient profiles near the surface. Therefore, three alternative assumptions are compared. The upward-gradient profiles could continue with a downward gradient

above 5 m, and we choose $R = 60 \%$ which is the maximum $R$ directly observed. The upward-gradient profiles could show constant concentration from 5 m up to the surface. And the upward-gradient profiles could continue with still upward gradient up to the surface. According to the assumptions above, expected distributions of bulk concentrations are then calculated for the three zones, and the total bias of emission estimates is calculated for the two cases of either using 10 m concentrations or 5 m concentrations instead of bulk concentrations (Table 1). Area weights are 0.5, 0.45, 0.05 for offshore, near-coastal, coastal

zone, respectively, because of their land distance ranges of 120 nm to 60 nm, 60 nm to 6 nm, 6 nm to 0 nm. The result is quite robust to the alternatives in the coastal zone, and to the choice of 10 m or 5 m concentrations: total emission bias $R$ is 20 to 25 % overestimation for the region encompassing all three zones. If confining the bias estimation to the near-coastal and coastal zones where gradients are found within the top 10 m, we can give a more general number for expected bias through near-surface gradients, as 20 % to 35 % overestimation. We see that the offshore zone has a low impact on bias due to the absence of an

$N_2O$ gradient on average and low $N_2O$ supersaturations causing low emission. The coastal zone has a low impact due to its small area and low wind speed causing low emission. The near-coastal zone with systematic downward gradients and moderate wind dominates the total bias like it dominates the total emission.

Note that this total bias is rather a conservative estimate, as we ignored extreme values of model runs and ship-based profiles, which suggest that downward gradients equivalent to $R > 100 \%$ may exist. Further we took into account the possibility that

profiles from the coastal zone with upward gradients might even continue with increasing concentration up to the surface.

## 5   Summary and conclusions

For the Peruvian upwelling region, we studied near-surface stratification and formation of near-surface gas gradients to obtain a consistent process picture of the air-sea gas exchange. We found that the peculiar setting composed of moderate wind conditions, subsequent near-surface stratification and surface trapping, in combination with strong local $N_2O$ production, lead to

the formation of strong and systematic near-surface $N_2O$ gradients. Observations combined with simple model calculations showed that the duration of near-surface stratification is the dominant influence on the strength of near-surface gas gradients. In particular the abundant multi-day near-surface stratification observed in the Peruvian upwelling region can explain the observed gas gradients, while near-surface stratification on diurnal or shorter timescales only has a minor impact. With the reported strong near-surface gas gradients, the sampling issue with the use of bulk flux parameterizations (Delta c issue)

is brought back to discussion, as the bias of inferred gas exchange seems non-negligible and is an order of magnitude larger compared to results obtained by Soloviev et al. (2002) for the open ocean. The impact on $N_2O$ emission estimates may even be of global relevance, because the global pattern of $N_2O$ high emission regions correlates with regions tending to surface trapping due to moderate wind. The Peruvian upwelling region alone is a large player in global oceanic $N_2O$ emissions,





with an estimated share of 5 to 20 % (Arévalo et al., 2015), and also other oceanic $N_2O$ hotspots like coastal and equatorial upwelling regimes may show favorable conditions for near-surface gradients. Arguably other uncertainties in gas exchange estimates may be equal or larger than the Delta c issue, e.g. transfer velocity parameterization uncertainties under low wind conditions (Garbe et al., 2014) or in the presence of surfactants (Tsai and Liu , 2003; Frew et al., 2004; Salter et al., 2011; Krall

et al., 2014). But the systematic bias in $N_2O$ emissions identified here can prospectively be eliminated by simpler means, be it by parameterization or changes in routine measurement strategy. So it deserves some effort to be understood better and be eliminated. As a result of this study, an 'educated screening' of the oceans for regions with expected strong near-surface gas gradients could enclose two criteria: near-surface stratification $N^2 \geq 10^{-4} \text{s}^{-2}$, and wind speed at 10 m between 3 and $6 \, \text{m s}^{-1}$. The findings also bring up open questions including what causes the extreme $N_2O$ near-surface distribution close to the coast.

High-resolution measurements here could help to clarify the existence, strength and conditions of a near-surface $N_2O$ source, and also add to better parameterize gas exchange at low wind conditions. The air-sea gas exchange of other gases might be affected in other ways by near-surface stratification. Gradients of photochemically produced substances with their main source near-surface may be much stronger than those of $N_2O$. Inferred fluxes of biologically active gases ($O_2$, $CO_2$) might even be altered in sign regionally.

*Data availability.* The used data sets are stored on the Kiel Ocean Science Information System (OSIS, https://portal.geomar.de/kdmi, data-management@geomar.de) and can be accessed upon request. According to the SFB 754 data policy (https://www.sfb754.de/de/data), all data associated with this publication will be published at a world data center (www.pangaea. de) when the paper is accepted and published. The $N_2O$ data presented here are archived in MEMENTO: https://memento.geomar.de/de

*Competing interests.* The authors declare that they have no conflict of interest.

*Acknowledgements.* We highly appreciate the support of the RV Meteor crew and scientific crew, and in particular thank Rudi Link and Andreas Pinck for constructing the near-surface water sampling equipment, Tina Baustian and Matthias Krüger who conducted oxygen and salinity measurements, and Gerd Krahmann for the final postprocessing of glider CTD and ship CTD data. The German Federal Ministry of Education and Research (BMBF) supported this study as part of the SOPRAN project (grant no. FKZ 03F0611A and 03F0662A). The German Science Foundation (DFG) provided support as part of Sonderforschungsbereich SFB754 "Climate Biogeochemistry Interactions in

the Tropical Ocean" and as part of cooperative project FOR1740. The daily wind fields from Metop/ASCAT scatterometer retrievals were obtained from the Centre de Recherche et d'Exploitation Satellitaire (CERSAT), at IFREMER, Plouzané (France). We thank the Peruvian authorities for permitting us to conduct the study in their territorial waters.



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



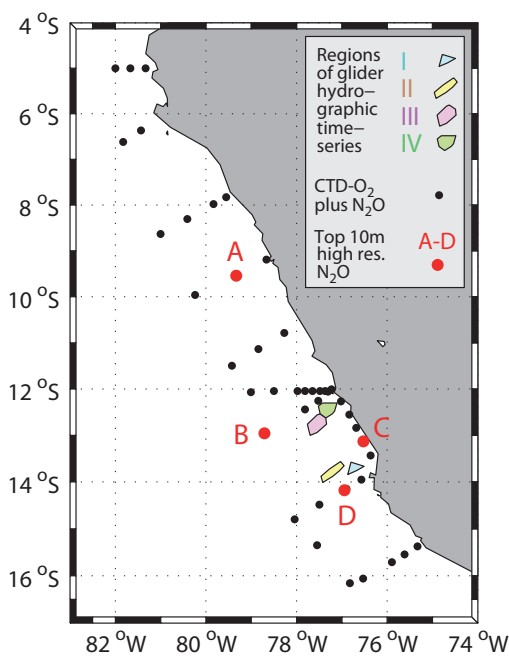

**Figure 1.** Locations of sample stations and glider time series off the coast of Peru, Dec. 2012 to Feb. 2013. Black dots: Simultaneous $CTD-O_2$ and $N_2O$ sampling, comprising 5 m and 10 m depth samples, during M91 (Dec. 3, 2012 – Dec. 23, 2012). Red dots: Zodiac based high-resolution $N_2O$ profiles of topmost 10 m; A: Dec. 8, 2012 16:30 local time; B: Dec. 13, 2012 10:00 local time; C: Dec. 16, 2012 14:30 local time; D: Dec. 17, 2012 14:00 local time. Colored areas: regions where time series of glider near-surface hydrography were obtained; I: 10 days from Feb. 17 to 27, 2013; II: 22 days from Jan. 23 to Feb. 22, 2013; III: 31 days from Jan. 15 to Feb. 15, 2013; IV: 37 days from Jan. 11 to Feb. 17, 2013.





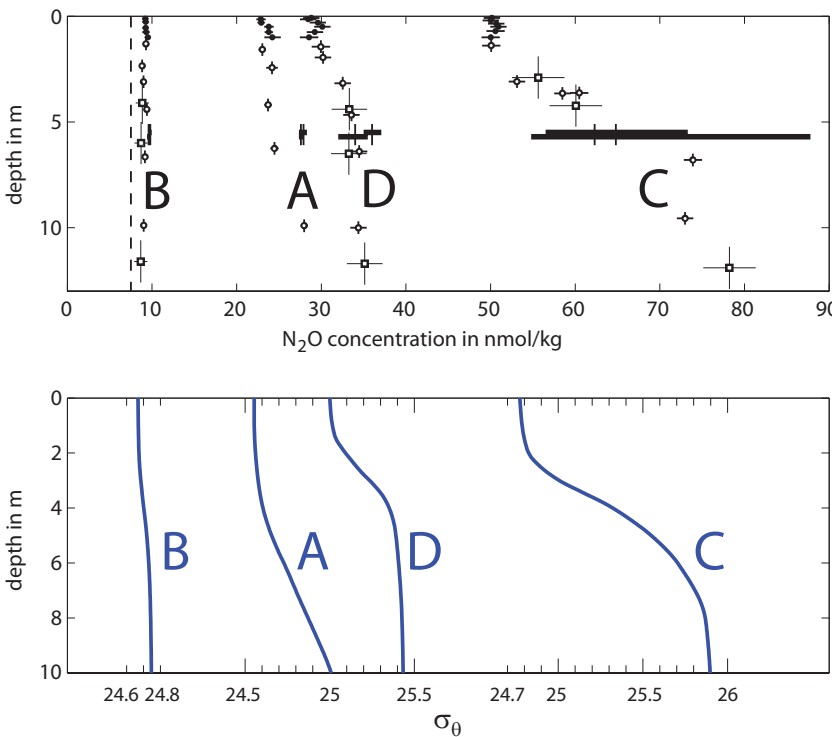

**Figure 2.** $N_2O$ and density profiles at the off-ship high-resolution stations A to D, complemented by shipboard observations at adjacent positions and times. For positions and times of station A to D cf. Fig. 1. Distances to land: B 106 nm, A 48 nm, D 36 nm, C 7 nm. 95%-limits of 10 m-wind distribution in $m\,s^{-1}$: B [4.3 6.6], A [3.1 6.0], D [2.6 5.9], C [3.3 5.4]. Upper panel: $N_2O$ measurements with 95 % confidence limits from measurement uncertainty; black dots in top 1 m: samples from centrifugal pump off-ship; black circles below 1 m: samples from Niskin bottle off-ship; black squares: samples from shipboard CTD; thick black lines: 95 % limits of distribution of ship underway samples during approach/leaving of the station, median values are marked. Lower panel: Density profiles derived from MicroCat temperature and conductivity profiles at station A to D.





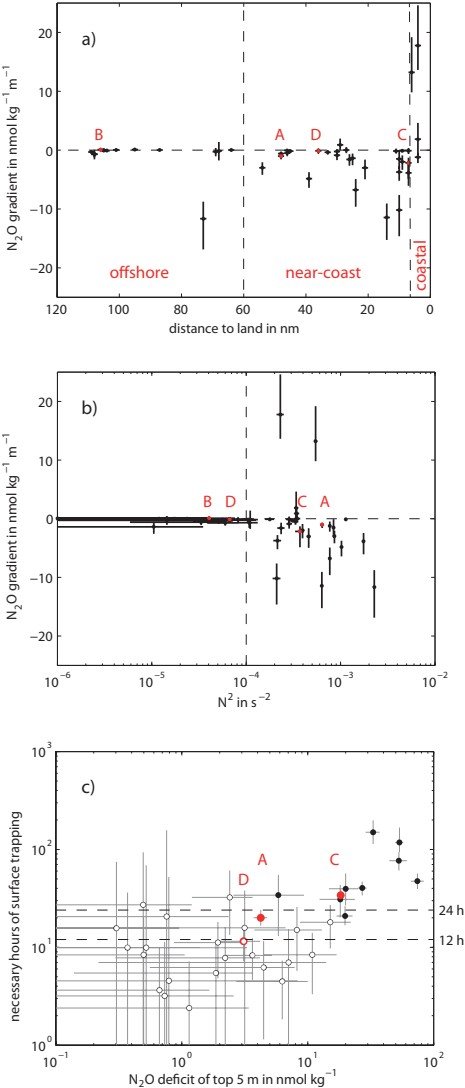

**Figure 3.** Characteristics of shallow $N_2O$ gradients derived from shipboard samples. The $N_2O$ gradient is calculated from bottle samples at about 5 m and about 10 m depth, negative gradients are defined as concentration decreasing with vertical coordinate z or increasing with depth. Error bars are 95 % confidence limits based on measurement uncertainty. Red symbols are high-resolution stations A to D (cf. Figs. 1 and 2). Top panel: $N_2O$ gradient vs. distance to land, calculated as shortest distance to coast. Dashed vertical lines separate three zones (offshore, near-coast, coastal) , dominated by neutral, downward, upward gradients, respectively. Centre panel: $N_2O$ gradient vs. buoyancy frequency squared, $N^2$, calculated from densities of the according $N_2O$ bottle samples. The dashed vertical line at $N^2 = 10^{-4} \, s^{-2}$ marks the approximate threshold below which no strong $N_2O$ gradients occur. Bottom panel: $N_2O$ deficit vs. estimated necessary time span of surface trapping, $N_2O$ deficit is concentration difference between 10 m and 5 m; hours of isolation are the time needed to deplete a 5 m water column from the 10 m-concentration down to the 5 m-concentration. Filled circles are stations where the necessary isolation time includes minimum one entire night, even for the lower confidence limit. Open circles are stations where night mixing cannot be excluded. Station B showed no negative gradient and is not part of the plot.



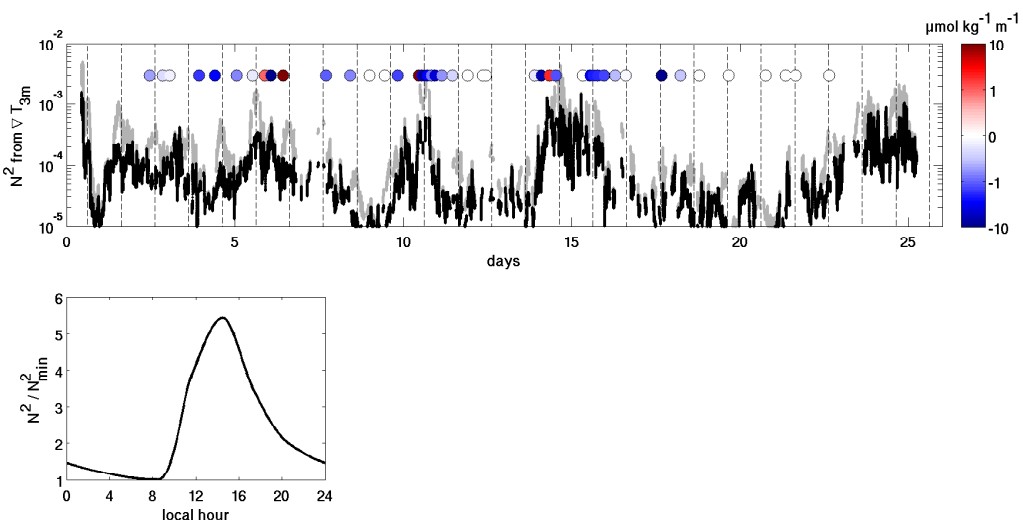

**Figure 4.** Upper panel: Observed near-surface $N_2O$ gradients vs. stratification at 3 m, in December 2012. Grey line: $N^2$ at 3 m estimated from hull temperature variance and ship motion variance; gaps are during stations. Diurnal periodicity is visible most days. Black line: same with mean diurnal cycle subtracted, by that mimicking the expected minimum nighttime stratification at each location. Colored dots: $N_2O$ gradient between about 10 m and 5 m depth from ship-based discrete sampling. Dashed lines mark 15:00h local time. Lower panel: Mean diurnal cycle of stratification, relative to minimum nighttime stratification $N_{min}^2$.





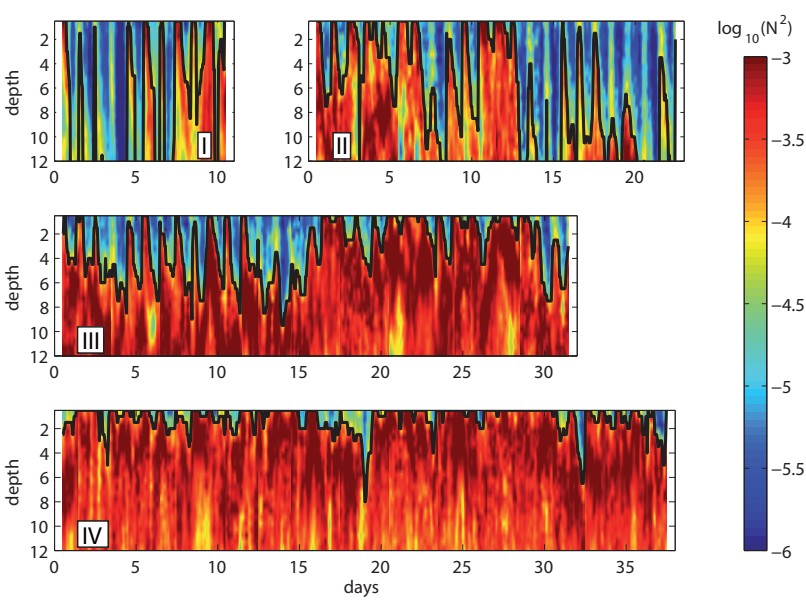

**Figure 5.** Near-surface stratification in composite glider hydrographic time series, sorted by increasing grade of persistence, from dominated by diurnal cycle to dominated by multi-day events. I, II, III, IV: regions of glider time series (Fig. 1). Black line: minimum depth of $N^2 \geq 10^{-4}\,\mathrm{s}^{-2}$, as base of the top layer (TLD, subsection 2.2.3). Time series are composites of different, partly overlapping glider sections in respective regions. $N^2$ processed in 0.5 m vertical bins, after low-pass-filtering the hydrographic time series (half power $k = \frac{1}{12}\,\mathrm{h}^{-1}$, cutoff $\frac{1}{3}\,\mathrm{h}^{-1}$) to eliminate spurious variations of TLD caused by internal wave vertical motions.




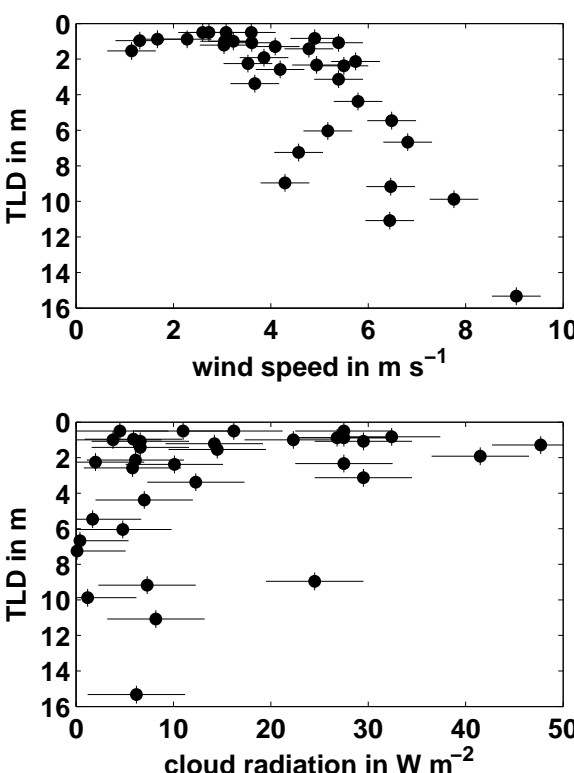

**Figure 6.** Influence of wind speed (upper panel) and cloud radiation (lower panel) on nighttime near-surface stratification. Night TLD is the night average from glider hydrographic time series (Fig. 5). Wind speed is the night rms average of ship wind from collocated positions (distance $\leq 0.3°$ lat/lon), converted to 10 m wind under non-neutral conditions using the COARE algorithm. Cloud radiation is the night average of long wave radiation minus clear sky long wave radiation.



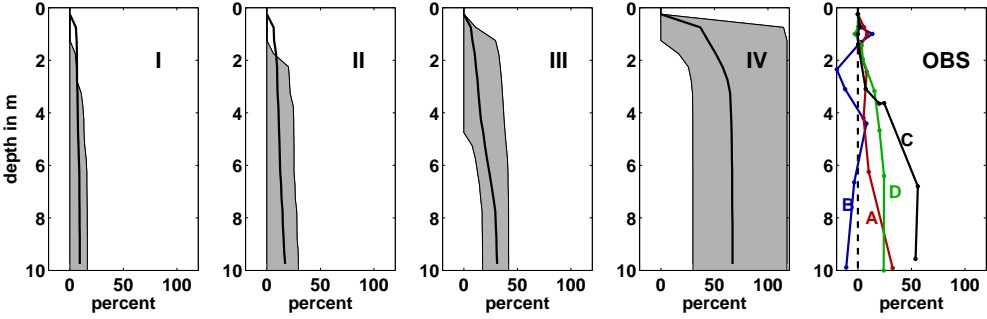

**Figure 7.** Modeled and observed $N_2O$ profiles, expressed as relative flux error R (subsection 2.2.6), i.e. equivalent to overestimation of air-sea gas exchange flux if using $N_2O$ at depth instead of bulk $N_2O$ in bulk flux parameterizations. I, II, III, IV: distributions of R in runs of 1-D transport model (subsection 2.2.7), forced by time series of TLD from respective glider time series, and by ASCAT wind speed. Thin lines/grey shading: 95 % limits of temporal distribution of flux overestimation at each depth. Thick lines: mean flux overestimation. OBS: flux overestimation of observed high-resolution profiles at sites A to D.

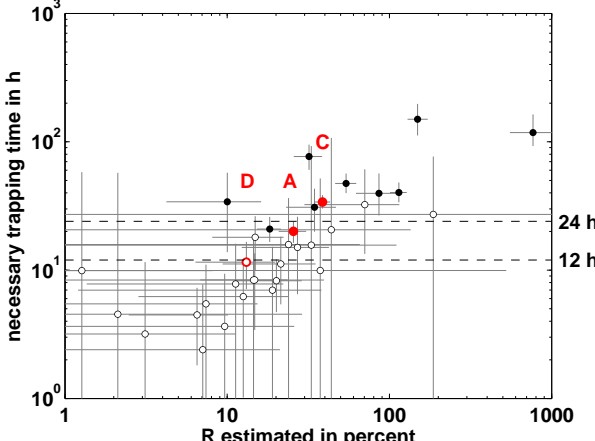

**Figure 8.** Necessary trapping time to explain observed differences between $N_2O$ concentrations at 5 m and 10 m, as a function of R. Assumed is depletion of the top 5 m layer by air-sea gas exchange due to observed wind. Due to the sparse resolution of $N_2O$ profiles at ship stations, R is estimated by setting $c_{bulk} = c_{5m}$ and $c_{ns} = c_{10m}$.




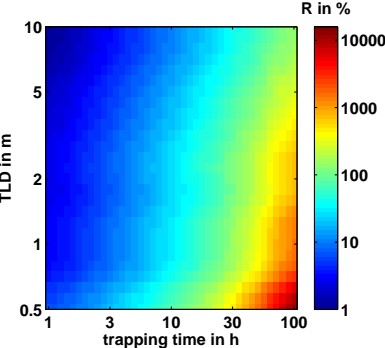
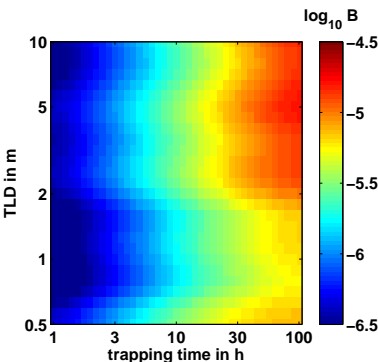

**Figure 9.** Gas exchange overestimation $R$ as a measure of relative gas exchange bias (left panel) and specific flux bias $B$ as a measure of absolute gas exchange bias (right panel), both as a function of trapping time $T_{trap}$ and top layer depth TLD. Based on corresponding values of wind speed $u_{10}$ and TLD as observed during the glider mission (Fig. 6), the field of $R(T_{trap}, TLD)$ has been interpolated and smoothed by a Gaussian algorithm. Assumed is a complete shutdown of $N_2O$ supply to the TL from below, and air-sea gas exchange transfer velocity following Nightingale et al. (2000).

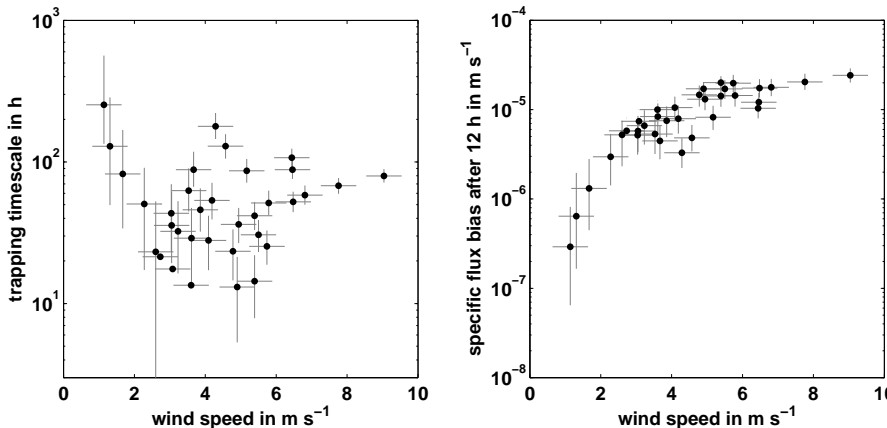

**Figure 10.** Trapping timescale $\frac{TLD}{k_w}$ (a measure of needed trapping to reach a certain R) as a function of wind speed $u_{10}$ (left panel), and specific flux bias $B = k_w \cdot \frac{R}{R+1}$ for R after 12 hours of trapping as a function of wind speed $u_{10}$ (right panel). The shape of $B(u_{10})$ is very robust to varying trapping time. R and B are based on the relation between wind speed $u_{10}$ and TLD as observed during the glider mission (Fig. 6). Assumed is a complete shutdown of $N_2O$ supply to the TL from below, and transfer velocity following Nightingale et al. (2000).





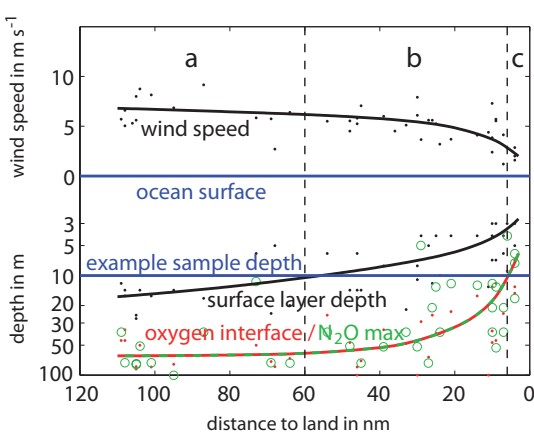

**Figure 11.** Observed distributions of wind speed, surface layer depth, oxygen interface depth, and depth of maximum $N_2O$ vs. distance to land in December 2012. Surface layer depth is an estimate of TLD from ship data (subsection 2.2.3). $N_2O$ max is the depth of shallowest local $N_2O$ maximum. Dots and circles are observations, lines represent schematic drawings. A constant sampling at 10 m (blue line) would intersect the TLD curve and the oxygen interface curve at two critical points with different distance to land (dashed vertical lines). The tilt of the layers leads to a perceived horizontal zonation of vertical $N_2O$ gradients (cf. Fig. 3a). a: offshore zone, b: near-coastal zone, c: coastal zone.





**Table 1.** Estimated average emission rates of $N_2O$ for December 2012 in different zones of the Peruvian upwelling region. Comparison between fluxes calculated from 10 m-, 5 m-, and surface (bulk) concentrations.

| $N_2O$ sea-to-air flux in $\mathrm{nmol\,m^{-2}\,s^{-1}}$ | Flux calculated from 10 m - concentrations | Flux calculated from bulk conc. (as derived from 10 m - conc.) | Flux calculated from 5 m - concentrations | Flux calculated from bulk conc. (as derived from 5 m - conc.) |
|---|---|---|---|---|
| Offshore zone 120 nm – 60 nm | 0.26 | 0.26 [0.24 0.29][1] | 0.14 | 0.14 [0.13 0.15][1] |
| Near-coastal zone 60 nm – 6 nm | 0.85 | 0.62 [0.58 0.67][2] | 0.68 | 0.53 [0.50 0.58][3] |
| Coastal zone 6 nm – 0 nm | 0.34 | 0.85[4]  0.61[5]  0.22[6] | 0.61 | 0.85[4]  0.61[5]  0.38[7] |
| All zones, area weighted average | 0.53 | 0.45[4]  0.44[5]  0.42[6] | 0.41 | 0.35[4]  0.34[5]  0.33[7] |
| Without offshore, area weighted average | 0.80 | 0.64[4]  0.62[5]  0.58[6] | 0.67 | 0.57[4]  0.54[5]  0.52[7] |

1  95 % confidence interval, based on the estimated range of flux overestimation in the offshore zone of -10 % to 10 %.

2  95 % confidence interval, based on the estimated range of flux overestimation in the near-coastal zone of 20 % to 60 % relative to 10 metres depth.

3  95 % confidence interval, based on the estimated range of flux overestimation in the near-coastal zone of 10 % to 50 % relative to 5 metres depth.

4  estimated surface concentration in the coastal zone is based on the assumption that the concentration gradient continues to the surface.

5  estimated surface concentration in the coastal zone is based on the assumption that the concentration is constant from 5 m upwards.

6  estimated surface concentration in the coastal zone is based on a flux overestimation of 60 % relative to 10 metres depth.

7  estimated surface concentration in the coastal zone is based on a flux overestimation of 60 % relative to 5 metres depth.