# Peer review of "Gas exchange estimates in the Peruvian upwelling regime biased by multi-day near-surface stratification"

_Biogeosciences, 2018_

## Referee Comment (RC1) · Anonymous Referee #1 · 20 Dec 2018

**1    General Comments**

The article by Fischer et al. is concerned with the impact of stratification on the air-sea gas exchange of $N_2O$, which leads to gradients of disolved nitrous oxide which diminish as the surface is approached.

This type of study has been carried out for $CO_2$, but this is the first time that such a study has been conducted for $N_2O$ in an upwelling region, which are recognised as hotspots for $N_2O$ emission. It is important to better constrain the air-sea exchange of $N_2O$ as it is thought that the ocean is a strong source of $N_2O$.

[Figure]

The authors provide reasonable and justifiable arguments for the effect of stratification on N2O gas exchange, and with some further effort the article could be published.

**2 Specific Comments**

P3L28: Add references to Sutherland et al. 2014 and Sutherland et al. 2016

For figure 4, can you add a mean diurnal cycle of temperature. I would like to see the extent of the thermal stratification.

section 2.2.5: can you add a histogram of the wind speed data used

section 2.2.7: I did not fully understand your model. Can you please elucidate with a schematic? I also did not understand why the model was constrained by the glider data only.

**3 Technical Corrections**

P5 L2: define OMZ

P5 L5: and will be called 'oxygen interface' in the following $->$ henceforth referred to as 'oxygen interface'

P5 L6: express 0.5 nm in meters

P5L7: ship-caused $->$ ship-induced

P7L27: Fig 4 comes before fig 2

P9L1: It is to be investigated $->$ Here we investigate

P10L18: you cannot start a sentence with I.e.

[Figure]

**4   References**

Sutherland, G., L. Marié, G. Reverdin, K. J. Christensen, G. Bröstrom, and B. Ward, 2016. Enhanced turbulence associated with the diurnal jet in the ocean surface boundary layer. J. Phys. Oceanogr., 46:3051–3067.

Sutherland, G., G. Reverdin, L. Marié, and B. Ward, 2014. Mixed and mixing layer depths in the ocean surface boundary layer under conditions of diurnal stratification. Geophys. Res. Lett., 41:8469–8476

---

## Referee Comment (RC2) · Jorge Vasquez (Referee) · 27 Feb 2019

The manuscript discusses the N20 gas exchange off the Peru Upwelling region. Overall the manuscript presents an important topic and I feel should be published after some revision. The primary conclusions is that the N2O gas-exchange Is dependent on the stratification and the gradients of N2O which are increasing downward (concentration less at the surface). The authors use data from gliders and shipboard CTD.

Major comments: perhaps I missed it but I did not see and connection the authors made with the seasonal upwelling cycle off Peru. Do they believe there are possible connections to trends in the upwelling cycle? Although the data limits the conclusions

on these time scales, can the authors, based on the results, speculate on the possible N2O relationship to longer trends, ENSO for example.

Minor comments: I believe the paper needs a thorough proofread before acceptance for publication. The overall quality of the figures I feel can also be improved. For example some of the figure had labeled "Day", but this is confusing. Day of what?
* * *

---

## Author Comment (AC1) · 27 Mar 2019

1 General Comments

Referee Comment:

The article by Fischer et al. is concerned with the impact of stratification on the air-sea gas exchange of N2O, which leads to gradients of disolved nitrous oxide which diminish as the surface is approached. This type of study has been carried out for CO2, but this is the first time that such a study has been conducted for N2O in an upwelling region, which are recognised as hotspots for N2O emission. It is important to better constrain

the air-sea exchange of N2O as it is thought that the ocean is a strong source of N2O. The authors provide reasonable and justifiable arguments for the effect of stratification on N2O gas exchange, and with some further effort the article could be published.

Answer:

Thank you very much for reviewing the manuscript and giving valuable advice. We now clarify the relation between existing research on CO2 exchange bias and our study, by complementing the Introduction after introducing the surface trapping (P4L2). In that context we also added a study by Miller et al. (2019), who conducted research in a similar way to our study, only for the Arctic ocean and CO2. Their paper was published after submission of our manuscript. The Introduction now reads from P4L3: 'For dissolved gases, vertical gradients in the top meters due to surface trapping had been predicted (McNeil and Merlivat, 1996), and later were indeed observed for oxygen and carbon dioxide (Soloviev et al., 2002; Calleja et al., 2013; Miller et al., 2019). Vertical concentration gradients due to surface trapping cause an additional bias in gas exchange estimates, independent of issues with solubility estimates which are caused by temperature gradients and which have particularly been studied to quantify CO2 exchange bias (e.g., Ward et al., 2004; Woolf et al, 2016). Concerning the surface trapping, the studies of Soloviev et al. (2002) and Calleja et al. (2013) showed that vertical concentration differences in oxygen and carbon dioxide exist across the top meters of several open ocean regions, however with little average effect on gas exchange estimates. Miller et al. (2019) found CO2 concentration gradients across the top meters of the Arctic ocean, and diagnosed substantial errors in CO2 exchange estimates if sampling below the surface layer. This may be rather a case of a very shallow seasonal mixed layer than a case of temporal surface trapping, but still underlines the practical importance of near-surface stratification and the DeltaC sampling issue. In coastal upwelling regions, there have been no reports on near-surface gas gradients so far. However, the conditions here for near-surface stratification and gradients should be more favorable than in the oligotrophic open ocean, ...'

2 Specific Comments

Referee Comment: P3L28: Add references to Sutherland et al. 2014 and Sutherland et al. 2016

Answer: Thank you for bringing these references to our attention. We add now references at the indicated place, now reading: 'Observations mainly from the open ocean revealed a diurnal cycle of near-surface temperature which is associated with the build-up of shallow stratification during daytime and its destruction during nighttime. This picture has become more and more detailed, as timeseries of high-resolution profiles in the undisturbed surface ocean have become available, from buoys (Prytherch et al., 2013; Wenegrat et al., 2015) and a free-rising profiler (Sutherland et al., 2014, 2016).' Sutherland et al. (2016) is now also added to P3L32 as a study with extensive observations.

Referee Comment: For figure 4, can you add a mean diurnal cycle of temperature. I would like to see the extent of the thermal stratification.

Answer: The observations of temperature at the thermosalinograph inlet during the cruise in December 2012 show a distinct diurnal cycle with an amplitude of 0.6 Kelvin at 3m depth. Unfortunately, there is no additional continuous record of temperature at other depths, which could have provided a more complete picture of the near-surface T profile in its dependence on external factors. We add a new panel to Fig.4 showing the mean diurnal T cycle at 3m depth.

Referee Comment: section 2.2.5: can you add a histogram of the wind speed data used

Answer: We add a histogram of rms averaged wind speed at N2O stations (averaged during 6h and in a radius of 5nm max.) as a new figure. The figure is now referred to in the text at P8L11, to illustrate that the encountered wind speed was in the low to medium range.

Referee Comment: section 2.2.7: I did not fully understand your model. Can you please elucidate with a schematic? I also did not understand why the model was constrained by the glider data only.

Answer: We add a schematic as a new figure, which incorporates on the one hand the terms we used in the text, and on the other hand the variables and flux equations we used for the model runs.

Fig.: The one-dimensional gas-transfer model to simulate the surface trapping mechanism. The interface of complete mixing inhibition shifts up and down according to the high-resolution timeseries of observed TLD without instantaneously affecting local N2O concentration. Vertical N2O transport is achieved by mixing within the two layers after the shifting interface has left a portion of clower-water in the top layer, or vice versa.

The model needs a high-resolution timeseries of undisturbed near-surface density profiles, in order to constrain it with a meaningful observational TLD-timeseries. Particularly the early morning hours are important, as destruction of the near-surface stratification can happen on short timescales having considerable effect on the N2O distribution. The glider data set we use is unique in that it consists of four timeseries of several weeks duration, showing undisturbed near-surface density profiles with multi-day stratification occurring. We are not aware of other suitable data sets in the Peruvian upwelling regime that would provide adequate TLD timeseries to use such a model to study the effects of multi-day near-surface stratification. We will augment the text of subsection 2.2.7, explaining the close interrelationship between the glider based TLD-timeseries and the 1-D model for studying the effects of multi-day near-surface stratification.

3 Technical Corrections

Referee Comment: P5 L2: define OMZ

Answer: The abbreviation 'OMZ' is now introduced in the Introduction P2L21 where the oxygen minimum zone of the tropical South Pacific is first mentioned explicitly.

Referee Comment: P5 L5: and will be called 'oxygen interface' in the following − > henceforth referred to as 'oxygen interface'

Answer: Changed to suggested wording

Referee Comment: P5 L6: express 0.5 nm in meters

Answer: Changed to 'at least 1km' instead of 'at least 0.5 nm'

Referee Comment: P5L7: ship-caused − > ship-induced

Answer: Changed to suggested wording

Referee Comment: P7L27: Fig 4 comes before fig 2

Answer: We removed the reference to Fig. 4 at this place, and rearranged the text to get the point clarified without a figure reference. It now reads from P7L25: 'However, horizontal temperature variability on short scales, vertical movements of the water column, and sensor noise add to temperature variance. The salinity required to convert the temperature gradient into stratification is taken from the thermosalinograph record, using the average salinity during the respective time bin, i.e. assuming a vertical salinity gradient of zero. After having calculated N2 at 3m depth for the entire cruise, we find an apparent lower limit of N2 > 10-5 s-2, which is probably caused by the temperature variance which is not due to the vertical temperature gradient.'

Referee Comment: P9L1: It is to be investigated − > Here we investigate

Answer: Changed to suggested wording

Referee Comment: P10L18: you cannot start a sentence with I.e.

Answer: Changed to 'So' instead of 'I.e.'.

4 References

Sutherland, G., L. Marié, G. Reverdin, K. J. Christensen, G. Bröstrom, and B. Ward, 2016. Enhanced turbulence associated with the diurnal jet in the ocean surface boundary layer. J. Phys. Oceanogr., 46:3051–3067.

Sutherland, G., G. Reverdin, L. Marié, and B. Ward, 2014. Mixed and mixing layer depths in the ocean surface boundary layer under conditions of diurnal stratification. Geophys. Res. Lett., 41:8469-8476

Added:

Miller L.A., Burgers, T.M., Burt, W.J., Granskog, M.A., and Papakyriakou, T.N. (2019): Air-sea $CO_2$ flux estimates in stratified Arctic coastal waters: How wrong can we be?, Geophysical Research Letters, 46, doi:https://doi.org/10.1029/2018GL080099.

Ward, B., Wanninkhof, R., McGillis, W.R., Jessup, A.T., DeGrandpre, M.D., Hare, J.E., and Edson, J.B. (2004): Biases in the air-sea flux of $CO_2$ resulting from ocean surface temperature gradients, Journal of Geophysical Research, 109, C08S08, doi:10.1029/2003JC001800.

Woolf, D.K., Land, P.E., Shutler, J.D., Goddijn-Murphy, L.M., and Donlon, C.J. (2016): On the calculation of air-sea fluxes of $CO_2$ in the presence of temperature and salinity gradients, Journal of Geophysical Research Oceans, 121, 1229-1248, doi: 10.1002/2015JC011427.

[Figure]

**Fig. 1.** Panel complementing Fig.4: Mean diurnal cycle of temperature at 3m depth

[Figure]

Fig. 2. Histogram of wind speed at the N2O stations

[Figure]

outgassing $\quad \Phi_{loss} = \rho k_w ( c_{top} - c_{eq})$

0 m

top layer $\quad c_{top}$

interface of
no mixing

TLD

lower layer $\quad c_{lower}$

12 m

supply by
diapycnal
mixing

$\Phi_{supply} = \rho K ( c_{20m} - c_{lower})/(20m - 0.5(TLD + 12m))$

**Fig. 3.** The one-dimensional gas-transfer model to simulate the surface trapping mechanism.

[Figure]

---

## Author Comment (AC2) · 27 Mar 2019

Referee Comment:

The manuscript discusses the N20 gas exchange off the Peru Upwelling region. Overall the manuscript presents an important topic and I feel should be published after some revision. The primary conclusions is that the $N_2O$ gas-exchange Is dependent on the stratification and the gradients of $N_2O$ which are increasing downward (concentration less at the surface). The authors use data from gliders and shipboard CTD.

Answer:

[Figure]

Thank you very much for reviewing the manuscript.

Referee Comment:

Major comments: perhaps I missed it but I did not see and connection the authors made with the seasonal upwelling cycle off Peru. Do they believe there are possible connections to trends in the upwelling cycle? Although the data limits the conclusions on these time scales, can the authors, based on the results, speculate on the possible N2O relationship to longer trends, ENSO for example.

Answer:

Indeed, we have no adequate observational data to directly answer to questions concerning seasonal variation or long-term trends. We now discuss our expectations for the typical seasonal cycle and for possible future developments of the upwelling system. This is done at the end of discussion section 4.4 for the two distinct aspects: 1) what may happen to N2O fluxes, 2) what may happen to the N2O flux bias due to the DeltaC sampling issue. 1) The seasonal and future amount of N2O emissions is indecisive, because it depends largely on the frequency and intensity of the mentioned peripheral hotspot N2O production, which is not fully understood and probably depends sensibly on all kinds of boundary conditions like the structure and dynamics of the oxygen field, mixing, or nutrient availability. The N2O emission during other seasons could in principle be accessed by measurement campaigns, but for future scenarios in coastal upwelling regimes, Capone and Hutchins (2013) state that the net effect on the state of the nitrogen cycle cannot be answered. As N2O formation is highly sensitive to the balance of nitrogen metabolic pathways, future N2O formation cannot be predicted. 2) The situation for the prediction of the emission bias is much better in our opinion, because the most important influence on the formation of near-surface gas gradients is the wind speed. Wind speed is higher than in December to February during the entire year (Echevin et al., 2008), and is expected to intensify in the future (Capone and Hutchins, 2013). We can use Fig. 10 (right panel) and Fig 6 (upper panel) to derive

the expectation that an intensified wind field should lead to a higher bias in emission estimates, but only if the wind speed does not exceed about 6m/s. In the latter case the trapping depth shifts to such depths that the typical sampling from the ship happens inside the trapping layer, i.e. without bias. So, the N2O flux bias may increase to some extent in regions of former low wind (Fig. 10), but the situations in which substantial bias occurs (wind speed 3 to 6 m/s) will become disproportionately rarer (Fig. 6), so that the net effect of an intensified wind field will probably be less bias.

Referee Comment:

Minor comments: I believe the paper needs a thorough proofread before acceptance for publication. The overall quality of the figures I feel can also be improved. For example some of the figure had labeled "Day", but this is confusing. Day of what?

Answer:

We agree that the paper will benefit from another round of proofreading, particularly to better stress the main points of the paper ( - gas gradients occur such shallow and grave that they are a substantial issue for routine gas exchange measurements; - the prominent role of the multi-day timescale of stratification for forming the gas gradients; - exploring the process and its impact) and concerning some parts that can be shortened. The time axis in Fig. 4 will be changed to a date axis, as it is just showing the time elapsed since start of the cruise in December 2012, and single days are easily identified by the daily dashed lines at 15 h local time. The time axes of Fig. 5 will be renamed to 'duration of hydrographic time series in days', because changing to date axes would in our opinion not add clarity here. Keeping the axes as elapsed days allows the reader to immediately see the duration of the time series as well as the duration of multi-day events. Instead, we add a sentence to the caption stating that all time series are from Jan/Feb 2013 and their exact dates can be obtained from the caption of Fig.1.

Added references

Echevin, V., Aumont, O., Ledesma, J., and Flores, G. (2008): The seasonal cycle of surface chlorophyll in the Peruvian upwelling system: A modelling study, Progress in Oceanography, 79, 167-176, doi: 10.1016/j.pocean.2008.10.026.

Capone, D.G. and Hutchins, D.A. (2013): Microbial biogeochemistry of coastal upwelling regimes in a changing ocean, nature geoscience, 6, 711-717, doi: 10.1038/NGEO1916.
* * *

---

## Author Response (AR1)

**Response to Referee comments on**

"Gas exchange estimates in the Peruvian upwelling regime biased by multi-day near-surface stratification" Fischer, T., Kock, A., Arévalo-Martínez, D.L., Dengler, M., Brandt, P., Bange, H.W.

**Referee #1**

**1** General Comments**

The article by Fischer et al. is concerned with the impact of stratification on the air-sea gas exchange of N2O, which leads to gradients of disolved nitrous oxide which diminish as the surface is approached.

This type of study has been carried out for CO2, but this is the first time that such a study has been conducted for N2O in an upwelling region, which are recognised as hotspots for N2O emission. It is important to better constrain the air-sea exchange of N2O as it is thought that the ocean is a strong source of N2O.

The authors provide reasonable and justifiable arguments for the effect of stratification on N2O gas exchange, and with some further effort the article could be published.

Thank you very much for reviewing the manuscript and giving valuable advice. We now clarify the relation between existing research on  $CO_2$  exchange bias and our study, by complementing the Introduction after introducing the surface trapping (P4L2). In that context we also added a study by Miller et al. (2019), who conducted research in a similar way to our study, only for the Arctic ocean and  $CO_2$ . Their paper was published after submission of our manuscript.

**The Introduction now reads from P4L3:**

'For dissolved gases, vertical gradients in the top meters due to surface trapping had been predicted (McNeil and Merlivat, 1996), and later were indeed observed for oxygen and carbon dioxide (Soloviev et al., 2002; Calleja et al., 2013; Miller et al., 2019). Vertical concentration gradients due to surface trapping cause an additional bias in gas exchange estimates, independent of issues with solubility estimates which are caused by temperature gradients and which have particularly been studied to quantify CO2 exchange bias (e.g., Ward et al., 2004; Woolf et al, 2016). Concerning the surface trapping, the studies of Soloviev et al. (2002) and Calleja et al. (2013) showed that vertical concentration differences in oxygen and carbon dioxide exist across the top meters of several open ocean regions, however with little average effect on gas exchange estimates. Miller et al. (2019) found CO2 concentration gradients across the top meters of the Arctic ocean, and diagnosed substantial errors in CO2 exchange estimates if sampling below the surface layer. This may be rather a case of a very shallow seasonal mixed layer than a case of temporal surface trapping, but still underlines the practical importance of near-surface stratification and the Delta c sampling issue. In coastal upwelling regions, there have been no reports on near-surface gas gradients so far. However, the conditions here for near-surface stratification and gradients should be more favorable than in the oligotrophic open ocean, ...'

**2** Specific Comments**

**P3L28: Add references to Sutherland et al. 2014 and Sutherland et al. 2016**

Thank you for bringing these references to our attention.

We add now references at the indicated place, now reading: 'Observations mainly from the open ocean revealed a diurnal cycle of near-surface temperature which is associated with the build-up of shallow stratification during daytime and its destruction during nighttime. This picture has become more and more detailed, as timeseries of high-resolution profiles in the undisturbed surface ocean have become available, from buoys (Prytherch et al., 2013; Wenegrat et al., 2015) and a free-rising profiler (Sutherland et al., 2014, 2016).' Sutherland et al. (2016) is now also added to P3L32 as a study with extensive observations.

**For figure 4, can you add a mean diurnal cycle of temperature. I would like to see the extent of the thermal stratification.**

The observations of temperature at the thermosalinograph inlet during the cruise in December 2012 show a distinct diurnal cycle with an amplitude of 0.6 Kelvin at 3m depth. Unfortunately, there is no additional continuous record of

temperature at other depths, which could have provided a more complete picture of the near-surface T profile in its dependence on external factors.

We add a new panel to Fig.4 showing the mean diurnal T cycle at 3m depth:

and a sentence to the results section motivating the additional panel: 'The presence of surface trapping is also revealed by the mean diurnal cycle of temperature at 3 m, with a mean amplitude of 0.6 K.'

**section 2.2.5: can you add a histogram of the wind speed data used**

We add a histogram of rms averaged wind speed at  $N_2O$  stations (averaged during 6h and in a radius of 5nm max.) as a new figure. The figure is now referred to in the text at P8L11, to illustrate that the encountered wind speed was in the low to medium range.

**section 2.2.7: I did not fully understand your model. Can you please elucidate with a schematic? I also did not understand why the model was constrained by the glider data only.**

We add a schematic as a new figure, which incorporates on the one hand the terms we used in the text, and on the other hand the variables and flux equations we used for the model runs. The figure also depicts the processing sequence during one model timestep to further illustrate how the model simulates gas transfer across the interface of no mixing.

Fig.: Panel a: The one-dimensional gas-transfer model to simulate the surface trapping mechanism. The interface of complete mixing inhibition shifts up and down according to the high-resolution timeseries of observed TLD without instantaneously affecting local  $N_2O$  concentration. Vertical  $N_2O$  transport is achieved by mixing within the two layers after the shifting interface has left a portion of  $c_{lower}$ -water in the top layer, or vice versa. Panels b1-b2-b3 demonstrate the processing sequence during a model timestep. b1: for the time of the timestep, supply flux and outgassing flux change  $c_{top}$  and  $c_{lower}$ , resulting in intermediate concentrations  $c_{top,i}$  and  $c_{lower,i}$ . b2: after the timestep, the TLD is shifted, in the example to a greater depth. b3: Instantaneous linear mixing within the new top and lower layers results in concentrations  $c_{top,1}$  and  $c_{lower,1}$ , which serve as start values for the next model timestep.

The model needs a high-resolution timeseries of undisturbed near-surface density profiles, in order to constrain it with a meaningful observational TLD-timeseries. Particularly the early morning hours are important, as destruction of the near-surface stratification can happen on short timescales having considerable effect on the  $N_2O$  distribution. The glider data set we use is unique in that it consists of four high-resolution timeseries of several weeks duration, showing undisturbed near-surface density profiles with multi-day stratification occurring. We are not aware of other suitable data sets in the Peruvian upwelling regime that would provide adequate TLD timeseries to use such a model to study the effects of multi-day near-surface stratification.

The text of subsection 2.2.7, P9L10 to P9L15, is now expanded to explain that the glider data meet the requirements to be the TLD time series constraining the model:

'That means that top and lower layer can change thicknesses, and entrain water of each other, which leads to the exchange of  $N_2O$  between the layers (Fig. 3, panels b1-b2-b3). For our purposes, the model needs to be constrained by realistic fluxes and high-resolution time series of TLD data, representative for the conditions in the Peruvian upwelling regime. Particularly the TLD time series require attention, as on the one hand locating the TLD needs undisturbed high-resolution information on the top meters of the water column, and on the other hand the temporal resolution must be fine enough to catch the principal TLD shifts through the hours of the day. Especially the expected TLD

maximum in the morning and the TLD minimum in the afternoon should be reliably resolved. We use observational data from 4 locations in the upwelling regime (region I, II, III, IV in Fig.1). The locations represent different grades of near-surface stratification, from domination by diurnal episodes to domination by multi-day events. The corresponding 4 time series of TLD are obtained from glider hydrographic near-surface profiles in January/February 2013 (cf. subsections 2.1, 2.2.2, and Thomsen et al.(2016)), as they represent undisturbed near-surface data of high spatiotemporal resolution.'

**3 Technical Corrections**

**P5 L2: define OMZ**

The abbreviation 'OMZ' is now introduced in the Introduction P2L21 where the oxygen minimum zone of the tropical South Pacific is first mentioned explicitly.

**P5 L5: and will be called 'oxygen interface' in the following - > henceforth referred to**

as 'oxygen interface'

Changed to suggested wording

**P5 L6: express 0.5 nm in meters**

Changed to 'at least 1km' instead of 'at least 0.5 nm'

**P5L7: ship-caused - > ship-induced**

Changed to suggested wording

**P7L27: Fig 4 comes before fig 2**

We removed the reference to Fig. 4 at this place, and rearranged the text to get the point clarified without a figure reference. It now reads from P7L25:

'However, horizontal temperature variability on short scales, vertical movements of the water column, and sensor noise add to temperature variance. The salinity required to convert the temperature gradient into stratification is taken from the thermosalinograph record, using the average salinity during the respective time bin, i.e. assuming a vertical salinity gradient of zero. After having calculated  $N^2$  at 3m depth for the entire cruise, we find an apparent lower limit for  $N^2$  of about  $10^{-5}$  s-2, which is probably caused by the temperature variance which is not due to the vertical temperature gradient.'

**P9L1: It is to be investigated - > Here we investigate**

Changed to suggested wording

**P10L18: you cannot start a sentence with I.e.**

Changed to 'So' instead of 'I.e.'.

**We add a paragraph at the end of subsection 4.4:**

'Our data base is representative only for December 2012 to February 2013, and also the  $N_2O$  field and emissions are not yet well constrained for other seasons and years. While the  $N_2O$  emissions at other seasons could principally be surveyed, predictions of a future trend remain largely uncertain. The latter is mainly due to partly competing effects of expected increased wind speed and expected increased stratification and leads to uncertain predictions of the future development of the nitrogen cycle in coastal upwelling regimes (Capone and Hutchins,2013), and the  $N_2O$ formation, which depends sensitively on complex boundary conditions. However, we may speculate about the seasonal and future bias associated with the Delta c sampling issue. We know that the wind speed during the entire year is higher than in December to February (Echevin et al.,2008), and it is expected to increase in the future (Capone and Hutchins,2013). Referring to our discussed results, the bias is most effective in the wind speed range between 3 and 6 m s-1 (Fig.12), and is practically absent beyond 6 m s-1, as the TLD will most probably lie below the sampling depth (Fig.8). When taking into account the observed wind distribution (Fig.6), and assuming a shift of the distribution to higher wind speeds, the wind range between 3 and 6 m s-1 will be less and less abundant, and the probability to find strong biases in the region will be reduced. So we expect less impact of the Delta c sampling issue for the seasons outside December to February, and also less impact in a future scenario characterized by increased winds.'

**Minor comments: I believe the paper needs a thorough proofread before acceptance for publication. The overall quality of the figures I feel can also be improved. For example some of the figure had labeled "Day", but this is confusing. Day of what?**

The summary part of the conclusions section has been reordered and complemented to better stress the main points of the paper ( - gas gradients occur such shallow and grave that they are a substantial issue for routine gas exchange measurements; - the prominent role of the multi-day timescale of stratification for forming the gas gradients). The time axis in Fig. 4 is now changed to a date axis, as it is just showing the time elapsed since start of the cruise in December 2012, and single days are easily identified by the daily dashed lines at 15 h local time. The time axes of Fig. 5 is renamed to 'duration of hydrographic time series in days', because changing to date axes would in our opinion not add clarity here. Keeping the axes as elapsed days allows the reader to immediately see the duration of the time series as well as the duration of multi-day events. Instead, we add a sentence to the caption stating that all time series are from Jan/Feb 2013 and their exact dates can be obtained from the caption of Fig.1.

[revised manuscript text omitted]